# ATTENTIVE SEQUENTIAL NEURAL PROCESSES

## ABSTRACT

Sequential Neural Processes (SNP) is a new class of models that temporally extends Neural Processes (NP) and can meta-learn a sequence of stochastic processes. This learned function however under-fits the provided contexts as is also the case in NP. Applying attention to the contexts resolves this in NP but simply extending this to SNP and applying attention on a buffer of context history is sub-optimal as our findings show. In this paper, we propose Attentive Sequential Neural Processes (ASNP) which resolves the under-fitting in SNP by introducing a novel imaginary context, modeled as a latent variable, over which the attention can then be applied. We evaluate our model on 1D Gaussian Processes regression and 2D moving MNIST/CelebA regression. We apply ASNP to implement Attentive Temporal-GQN and further evaluate on the moving CelebA task.

## 1 INTRODUCTION

Neural Processes (NP) (Garnelo et al., 2018b) combine the strengths of neural networks and Gaussian processes (GP) such that, like Gaussian processes, it is flexible in learning a new stochastic process at test time and also provides fast $\mathcal{O}(1)$ prediction speed like neural networks. Learning from small datasets of multiple tasks (i.e., multiple stochastic processes), NP can be seen as a probabilistic latent variable framework for meta-learning. Sequential Neural Processes (SNP) (Singh et al., 2019) extend the power of NP to a sequence of stochastic processes thus introducing a new class of sequential latent generative models. It targets a large class of problems where the sequence of stochastic processes is governed by underlying transition dynamics and where learning to transfer information between stochastic processes is useful.

Despite these strengths, the remained question is whether NP and SNP can scale to more complex stochastic processes when the individual observations about the true process are highly partial, which prevails in many situations. For example, an agent in a large complex landscape can only obtain very limited information from its immediate surroundings and it is desirable that it still maintains an accurate global scene representation that could also be changing with time. Studying NP from this perspective, Kim et al. (2019) demonstrates that the *under-fitting* problem significantly deteriorates the performance of NP. To address this, ANP (Kim et al., 2019) and the method in Rosenbaum et al. (2018) augment attention to NP and GQN (Eslami et al., 2018), respectively.

Given the severe performance degradation of NP due to the under-fitting, an important key question is *whether SNP would also suffer from the under-fitting problem or not*, and its follow-up questions such as *if so, how severely and in which settings would it happen?* and *how can we resolve the problem—would the same attention mechanism that worked for NP work for SNP as well?* In this paper, our goal is to answer these questions. We achieve this by not only showing that our proposed attention for SNP significantly improves the standard SNP under the under-fitting settings (see Fig. 1), but also by making a stronger claim i.e., in a sequential setting like ours, it is not optimal to perform attention on a memory buffer that simply stores all the observed contexts. Instead, through our experiments, we claim that attention should be performed on a memory that is also updated (learned) at every time-step. Consequently, *a)* this memory would learn to store an optimized representation of the past geared towards prediction and *b)* it would require fewer storage locations as it does not need to naively store every observed context points.

To implement such a memory, we introduce the concept of *imaginary context* that augments the set of real context points observed at any time-step. The imaginary context contains a fixed number of context points and is modeled as a random variable sampled at every time-step. Given a query, we then apply attention on the union of the real and the imaginary contexts to generate predictions. We call the proposed model as Attentive Sequential Neural Processes (ASNP). In experiments on

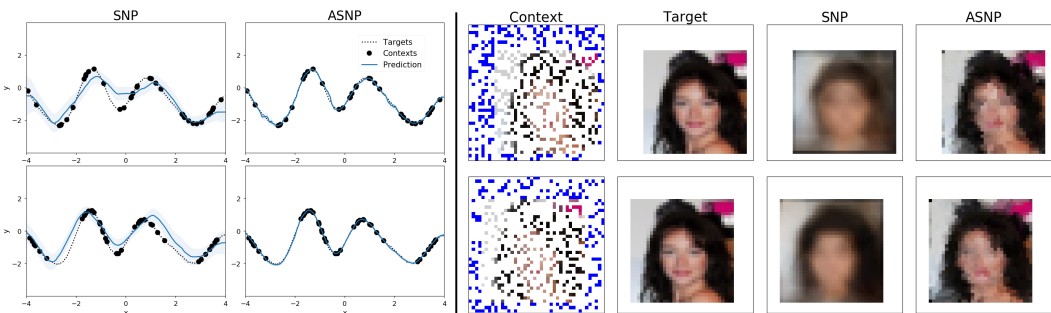

**Figure 1:** Demonstration of under-fitting in SNP by comparing against ASNP for 1D Gaussian Process and 2D moving CelebA regression tasks. Samples in the first and second rows are from $t = 0$ and $4$. In the context column for CelebA, white pixels for the background are drawn as blue for visualization.

1D regression tasks, we validate that SNP which simply attends on the memory buffer of all the past context points under-performs the proposed ASNP endowed with imaginary contexts. Then in a comprehensive set of experiments on 1D and 2D regression and rendering tasks, we demonstrate ASNP's performance gains over NP, ANP, SNP, GQN, and TGQN in different context regimes.

Our main contributions are as follows. *a)* We show that SNP suffers from significant under-fitting by testing it in settings when the true stochastic process is complex (e.g., complex CelebA faces moving on a white canvas) or when each observation is highly partial (e.g., each context point is just one pixel). *b)* We introduce a new model, Attentive Sequential Neural Processes (ASNP), that resolves the problem of under-fitting by augmenting SNP with an attention mechanism. To implement this, we introduce the novel concept of *imaginary context*. We also empirically demonstrate that SNP which simply performs attention on a lossless memory of all the past context points still under-performs the proposed model and therefore the use of imaginary context is a better design choice. *c)* Using the attention paradigm introduced in ASNP, we extend TGQN to Attentive TGQN (ATGQN) for scene rendering. *d)* We show that ASNP outperforms the baselines in various 1D and 2D tasks performing future prediction and meta-transfer learning i.e., making simultaneous use of the past knowledge and the current contexts.

## 2 BACKGROUND

**Neural Processes (NP)** (Garnelo et al., 2018b) learns to learn a stochastic process $\mathcal{P}$ that maps an input $x \in \mathbb{R}^{d_x}$ to a random variable $y \in \mathbb{R}^{d_y}$. It does so using a set of context observations $C = (X_C, Y_C) = \{x_i, y_i\}_{i \in \mathcal{I}(C)}$. Here, $\mathcal{I}(C)$ is the set of indices for the elements in set $C$. Specifically, to learn stochastic process from the context, NP uses an inference distribution $P(z|C)$ and then uses the representation $z \sim P(z|C)$ of the inferred stochastic process to generate the prediction $y$ given a query $x$. This is modeled by $p(y|x, z)$. The full generative process of NP conditioned on the context can be written as:

$$P(Y|X, C) = \int P(Y|X, z)P(z|C)\mathrm{d}z \tag{1}$$

where $P(Y|X, z) = \prod_{i \in \mathcal{I}(D)} P(y_i|x_i, z)$ and $D$ contains the *target* observations $D = (X, Y) = \{x_i, y_i\}_{i \in \mathcal{I}(D)}$. The training data for NP is obtained by drawing a stochastic process and then sampling $(C, D)$ from the sampled stochastic process. Treating each stochastic process as a task, NP can be seen as a probabilistic meta-learning framework.

**Generative Query Networks (GQN)** (Eslami et al., 2018) uses modeling similar to NP for the problem of learning representation and rendering of 3D scenes from a set of partial 2D observations. In GQN, a query $x$ becomes a camera viewpoint in a 3D environment and an output $y$ corresponds to an image taken from the viewpoint $x$. The latent variable in the original GQN are query-dependent and may thus produce uncorrelated generations across queries. Consistent GQN (CGQN) (Kumar et al., 2018) resolves this problem by having a single stochastic scene representation. For simplicity, we use the term *GQN* to denote CGQN in the remainder of the paper.

**Attentive Neural Processes (ANP)** (Kim et al., 2019) resolves the problem of under-fitting in NP. To this end, ANP uses two context-encoding paths. The first path is the same as the scene encoder in NP that summarizes all the context points agnostic to the target query $x$ and produces a single vector

of the global representation by average pooling. The other path computes representations for each target query using query-dependent attention. The set of representations returned $r$ from the set of target queries $X$ may be written as $r = f_{\text{attend}}(X, X_C, v_C)$, where $v_C$ are the encoded representations of $C$ via MLP or self-attention. The attention function $f_{attend}$ is a key-value based attention which computes a response for each query $q_j \in X$ by first computing its similarity with the key set $X_C$ as $w_{ji} = f_w(k_i, q_j)$ and then performing a weighted-sum on the value set i.e., $s_j = \sum_i w_{ji} v_i,$. Here, the keys and values are the context queries $x_C$ and the representations $v_C$, respectively. The generative process is as follows: $P(Y|X, C) = \int P(Y|X, z, f_{\text{attend}}(X, X_C, v_C))P(z|C)\mathrm{d}z$. Kim et al. (2019) propose ANP as a general model class which may be realized using different attention mechanism choices such as Dot-Product, Laplace or Multi-Head (Vaswani et al., 2017). ANP needs more computation than NP but Kim et al. (2019) mitigates this by showing faster training convergence than NP.

**Sequential Neural Processes (SNP)** (Singh et al., 2019) extend the capabilities of NP to model stochastic processes which may be decomposed as a sequence of stochastic processes $\mathcal{P}_1, \mathcal{P}_2, \cdots \mathcal{P}_T$ such that $\mathcal{P}_t$ at time $t$ depends on the past processes $\mathcal{P}_{<t}$ through some underlying temporal dynamics. At any $t$, $\mathcal{P}_t$ is modeled as $z_t$ which depends on the current context $C_t$ and the past processes $z_{<t}$ using the transition model $p(z_t|z_{<t}, C_t)$. This enables us to use the general past trend as well as the current context to meta-transfer learn the process $\mathcal{P}_t$. Given $z_t$ and a query $x_t$, the prediction $y_t$ is generated using the observation model $p(y_t|x_t, z_t)$. In the SNP framework, at time-step $t$, we cannot directly access the past contexts, but we can still directly access the current context $C_t$. Thus, to model $\mathcal{P}_t$, we can use the stochastic encoding $z_t$ together with the direct encoding of $C_t$. Alternatively, this allows us to write the observation model as $p(y_t|x_t, z_t, C_t)$. Combining the transition and the observation models, the complete generative process of SNP may be written as follows.

$$P(Y, Z|X, C) = \prod_{t=1}^{T} p(y_t|x_t, z_t, C_t)p(z_t|z_{<t}, C_t). \tag{2}$$

## 3 ATTENTIVE SEQUENTIAL NEURAL PROCESSES

In this section, we describe the proposed Attentive Sequential Neural Processes (ASNP) that augments SNP with an attention mechanism.

### 3.1 GENERATIVE PROCESS

We recall that ANP uses the observation model $p(y|x, z, C)$ which, based on the query $x$, performs attention on the context points in $C$. In SNP, which uses the observation model $p(y_t|x_t, z_t, C_t)$, an analogous approach would use $x_t$-dependent attention on $C_t$. However with this approach, attention is not effective because the current context is likely to be limited or even empty. Instead, we want to perform attention not only on the current but also on the past contexts. But to do that, we found that simply attending on a memory buffer of context history does not produce enough gains and leaves room for further improvement (see Appendix D.3). Instead, our hypothesis is that attention should be performed on a sequentially-updated memory so that the representations are optimized for prediction and are more size-efficient. To realize this, we propose augmenting the set of real

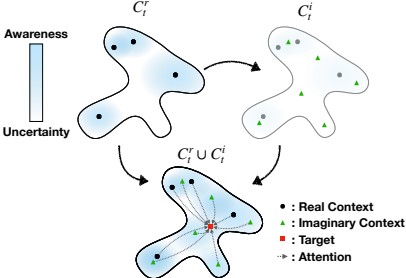

**Figure 2:** Illustration of imaginary context. Conditioned on real context, the imaginary context is generated which reduces the uncertainty at important points in the query space.

context points $C_t^r$ with a set of *imaginary* context points $C_t^i$ resulting in the full context set $C_t \equiv C_t^r \cup C_t^i$ so that the observation model can now perform attention on this augmented $C_t$. Here, $C_t^i = (X_t^i, U_t^i)$ consists of imaginary queries (or keys) $X_t^i = \{x_{t,k}^i\}_{k=1}^{K}$ and its corresponding values $U_t^i = \{u_{t,k}^i\}_{k=1}^{K}$ where $K$ is a hyperparameter similar to the number of slots in a memory. To generate $C_t^i$, we introduce a novel and attentive transition model $P(C_t^i|C_{<t}, C_t^r)$ treating $C_t^i$ as a random variable. In this way, $C_t^i$ is a noisy reconstruction of the past sampled using an attention mechanism on the past contexts (real and imaginary) and on the real contexts of the current

time-step. Augmenting SNP with these, the generative model for ASNP may be written as follows.

$$P(Y, Z, C^i | X, C^r) = \prod_{t=1}^{T} P(y_t | x_t, z_t, C_t) P(z_t | z_{<t}, C_t) P(C_t^i | C_{<t}, C_t^r).$$  (3)

**Implementing the transition model** $P(z_t | z_{<t}, C_t)$. With this model, we obtain the global representation $z_t$ which is a query-agnostic encoding of the current stochastic process. This module can be considered similar to its counterpart in SNP without attention. Note that it is a design choice whether to choose $C_t$ or $C_t^r$ for the conditioning context of this module. If we use $C_t^r$, $z_t$ becomes global context-encoding of real contexts. Otherwise, it becomes an encoding of the combined contexts.

**Implementing the context imagination** $P(C_t^i | C_{<t}, C_t^r)$. To implement this, we keep track of the $K$ imaginary context points at all time-steps and their generative model is as follows:

$$P(C_t^i | C_{<t}, C_t^r) = \prod_{k=1}^{K} P(x_{t,k}^i, u_{t,k}^i | C_{<t}, C_t^r) = \prod_{k=1}^{K} P(u_{t,k}^i | x_{t,k}^i, C_{<t}, C_t^r) P(x_{t,k}^i | C_{<t}, C_t^r).$$  (4)

For each imagination slot $k$, we first sample the key $x_{t,k}^i$. For this, we use an RNN that takes the past imaginary inputs $X_{t-1}^i$ and the encoding of the real contexts $v_t^r = f_{\text{order-inv}}(C_t^r)$ to produce a hidden state $h_t^X$ from which we then sample the imaginary key set $X_t^i$. Given these keys, the corresponding imaginary values $u_{t,k}^i$ are generated next. For each imagination slot $k$, we maintain an imagination-tracker RNN that takes $(x_{t,k}^i, u_{t-1,k}^i)$ as input and produces a hidden state $h_{t,k}^u$. These hidden states act as intermediate imaginary values and we use

| **Algorithm 1** Attention with Imagination |
| --- |
| $h_t^X = \text{RNN}_X((X_{t-1}^i, v_t^r), h_{t-1}^X)$ |
| $X_t^i \sim \mathcal{N}(f_\mu^X(h_t^X), f_\sigma^X(h_t^X))$ |
| $h_{t,k}^u = \text{RNN}_u((x_{t,k}^i, u_{t-1,k}^i), h_{t-1,k}^u)$ |
| $a_{t,k}^i = f_{\text{attend}}(x_{t,k}^i, S_t^i)$ |
| $u_{t,k}^i \sim \mathcal{N}(f_\mu^u(a_{t,k}^i), f_\sigma^u(a_{t,k}^i))$ |
| $r_{x_t} = f_{\text{attend}}(x_t, C_t)$ |

them to construct a key-value set $S_t^i = \{(x_{t,k}^i, h_{t,k}^u)\} \cup \{(x_t^r, f_{y \to u}(x_t^r, y_t^r))\}$. On this set, we apply self-attention using only the imaginary $x_{t,k}^i$ as attention queries and obtain the corresponding attention encodings $a_{t,k}^i$. Such encodings therefore also model the interactions among the context points. From these, we sample the final imagination values $u_{t,k}^i \sim \mathcal{N}(f_\mu^u(a_{t,k}^i), f_\sigma^u(a_{t,k}^i))$.

**Implementing the observation model** $P(y_t | x_t, z_t, C_t)$. Taking $x_t$ as the attention query, we attend on $C_t$, the union of the real $C_t^r$ and imagined $C_t^i$ context, as the key-value set and obtain the attention encoding $r_{x_t} = f_{\text{attend}}(x_t, C_t)$. This is done after each real context $x \in X_t^r$ and $y \in Y_t^r$ is encoded using $f_{y \to u}$. The attention encoding $r_{x_t}$ is then concatenated with the global encoding $z_t$ to perform the prediction $y_t$. Algorithm 1 shows the described process of context imagination and encoding.

**Discussion.** An interesting interpretation of the proposed modeling can be drawn in comparison to the memory retrieval process in the human brain (Eichenbaum, 2017). As is well-known in neuroscience, "*human memory is not a literal reproduction of the past, but instead relies on constructive processes that are sometimes prone to error and distortion*" (Schacter, 2012). Our model resembles this process in the sense that it (i) compresses the past observation and importantly what it has imagined in the past through an RNN (similarly to memory consolidation), and then after observing current context, (ii) recalls from the compressed memory not a simple copy of the past but a constructive recreation of representations that is optimized to help prediction as illustrated in Fig. 2.

### 3.2 LEARNING AND INFERENCE

Due to the intractability of the true posterior, ASNP is trained via variational approximation with the following posterior approximation:

$$P(Z, C^i | C^r, D) \approx \prod_{t=1}^{T} Q(z_t | z_{<t}, C_t^i, C^r, D) Q(C_t^i | C_{<t}^i, C^r, D)$$  (5)

where $Q(C_t^i | C_{<t}^i, C^r, D) = Q(U_t^i | X_t^i, C_{<t}^i, C^r, D) Q(X_t^i | C_{<t}^i, C^r, D)$ and $D = (X, Y)$. For training, the following evidence lower bound (ELBO) is maximized w.r.t. $\theta$ and $\phi$:

$$\mathcal{L}_{\text{ASNP}}(\theta, \phi) = \sum_{t=1}^{T} \mathbb{E}_{Q_\phi(z_t, C_t^i | C^r, D)} \left[ \log P_\theta(y_t | x_t, z_t, C_t^i, C_t^r) \right]$$

$$- \mathbb{E}_{Q_\phi(z_{<t}, C_{<t}^i)} \left[ \mathbb{KL} \left( Q_\phi(z_t, C_t^i | C^r, D) \parallel P_\theta(z_t, C_t^i | z_{<t}, C_{<t}^i, C_t^r) \right) \right].$$  (6)

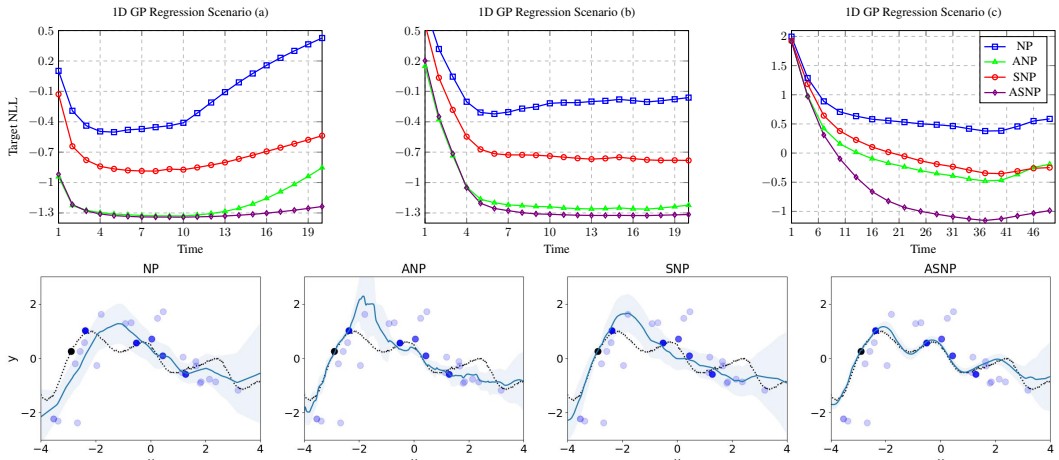

**Figure 3: Top:** Negative log-likelihood (NLL) for target points at each time-step for 1D regression. **Bottom:** Samples for 1D regression task (c) at $t = 33$. *Dotted line:* True function. *Blue line:* Predicted function. *Shaded light-blue region:* Uncertainty. *Black dot:* Current context points at $t = 33$. *Blue dots:* Past contexts where color intensity reflects the recentness.

For backpropagation, we use reparameterization trick (Kingma & Welling, 2013) for continuous variables. Derivation of eq. 6 is provided in Appendix A.

## 4 RELATED WORKS

Learning to learn stochastic processes from few data points at test time has become attractive as meta-learning in recent times. In this line of work, Conditional Neural Processes (CNP) (Garnelo et al., 2018a) models a regression function but without a global latent which causes inconsistency between values for different queries. NP (Garnelo et al., 2018b) resolves this by proposing an explicit global latent path. While CNP and NP are frameworks demonstrated on simpler tasks, GQN (Eslami et al., 2018) develops it to render 3D scenes. Like CNP, GQN also produces inconsistency between queries due to the absence of a global latent. CGQN (Kumar et al., 2018) resolves it by introducing a global scene latent. To address the challenge of scalability to larger and more complex stochastic processes, ANP (Kim et al., 2019) identifies and resolves the problem of under-fitting using query-dependent attention. Similarly, Rosenbaum et al. (2018) introduces attention into the GQN model to render complex 3D scenes in large procedurally-generated maps as in Minecraft. To extend the model to stochastic processes decomposable as a sequence of stochastic processes, SNP (Singh et al., 2019) was proposed. It was then used to extend GQN to TGQN for rendering dynamic 3D scenes. Functional Neural Processes (FNP) (Louizos et al., 2019) was proposed to learn a graph of dependencies between a pre-selected set of points and the training points for modeling distributions over functions.

The concept of imaginary context i.e. pseudo-context, which act as trainable representative points is related to the inducing points in Set Transformer (Lee et al., 2019) and sparse GPs (Snelson & Ghahramani, 2006; Titsias, 2009). VampPrior (Tomczak & Welling, 2017) also uses few trainable pseudo-inputs to resolve over-fitting, over-regularization and high computational complexity. The analogous reference set used in FNP to model a graph of small number of points is also motivated from the inducing points of sparse GPs. With regard to the use of trainable memory, Differentiable Neural Dictionary (Pritzel et al., 2017) stores and updates the previous trajectories as key-value pairs to quickly learn a variety of environments.

## 5 EXPERIMENTS

We evaluate ASNP on the following selection of tasks: *a)* 1D Gaussian Process (GP) regression *b)* 2D moving MNIST (LeCun et al., 1998) and 2D moving CelebA (Liu et al., 2015). On these tasks, we evaluate and compare the proposed model against NP, ANP, SNP as the baselines. Each task is evaluated with three scenarios, which is used in (Singh et al., 2019) for 1D GP regression: (a) To evaluate the model's ability to extrapolate the future, we provide high-amount of context

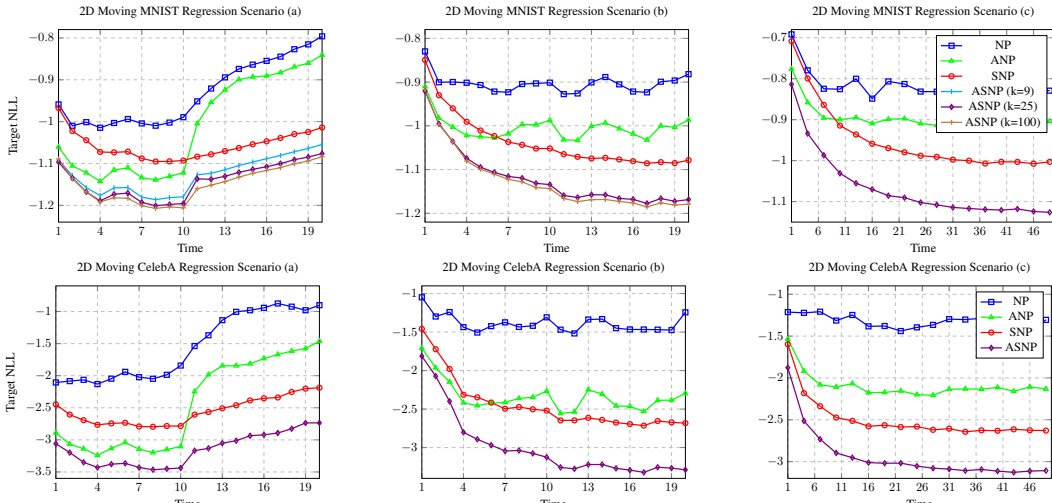

**Figure 4:** Target NLL at each time-step for 2D moving MNIST (**Top**) and CelebA (**Bottom**) regression.

in the first 10 time-steps out of the 20. (b) To test the model on its ability to track the dynamics using intermittently arriving context, we provide high-amount of context in 10 randomly chosen time-steps out of the 20. (c) To test the model's ability to gather and make use of low amount of context received over long segments of time, we provide low amount of context in 45 randomly chosen time-steps out of the 50. In each setting, the remaining time-steps are used for prediction. The details about model architectures can be found in Appendix B.1.

To test the benefits of attention for rendering 2D images, we used moving CelebA because it has higher uncertainty when given partial knowledge in comparison to an alternative data set such as moving simple shapes (e.g. circle). We test on the scenario (c) as described above, but with shorter sequence lengths and smaller context sizes. Details are explained in the section on 2d rendering.

**1D Regression:** We first evaluated our method on sequential 1D Gaussian Process data set. This is an extension of the GP data set through the addition of linear dynamics on the kernel hyper-parameters as used in (Singh et al., 2019). In each transition, we add a small Gaussian noise to introduce stochasticity. For scenarios (a) and (b), the number of context observations $n$ is randomly selected in $[5, 50]$ or an empty set. The number of target observations $m$ is randomly selected in $[1, 51 - n]$. For scenario (c), $n$ is 1 or 0, and $m$ is chosen from $[1, 11 - n]$. The target set subsumes the context. The more details about this task is described in Appendix C.

In the quantitative results of Fig 3, SNP outperforms NP as reported in (Singh et al., 2019). On the other hand, in scenario (a), the performance of ANP degrades steeply after the context are removed since it does not model the dynamics explicitly. Even though it appears to outperform NP and SNP, this can be credited to the attention in ANP that prevents underfitting (Kim et al., 2019). ASNP shows better performance for all the scenarios, outperforming ANP with a greater gap in scenario (c). Further, it saturates the fastest among the tested methods for the entire scenarios as described in Appendix D.1. We hypothesize that when the dense context is given at each time-step in (a) and (b), ANP can fit well as it does not need to rely on the dynamics much. But in scenario (c), the context is low in each time-step and the model must capture dynamics to make the best use of the context seen thus far.

In the qualitative results of Fig 3, we illustrate a sample of a validation set in scenario (c) with predictions from ASNP and the baselines. As shown, even for the points not shown recently, ASNP predicts better than the baselines with lower uncertainty. ANP and SNP fail to predict those points well. This shows that the update of the imaginary queries and the representations is necessary over and above the sequential update of the global representations or the expensive modeling of the entire past using the time information. More pieces of evidence about a useful role of imaginary context are shown in Appendix D.2 and D.3 where comparisons with larger size baselines and ASNP without imaginary context are described and ASNP outperforms them. It also shows a having more parameters in baselines is not sufficient and imaginary context plays a significant role. More qualitative results are in Appendix E.1.

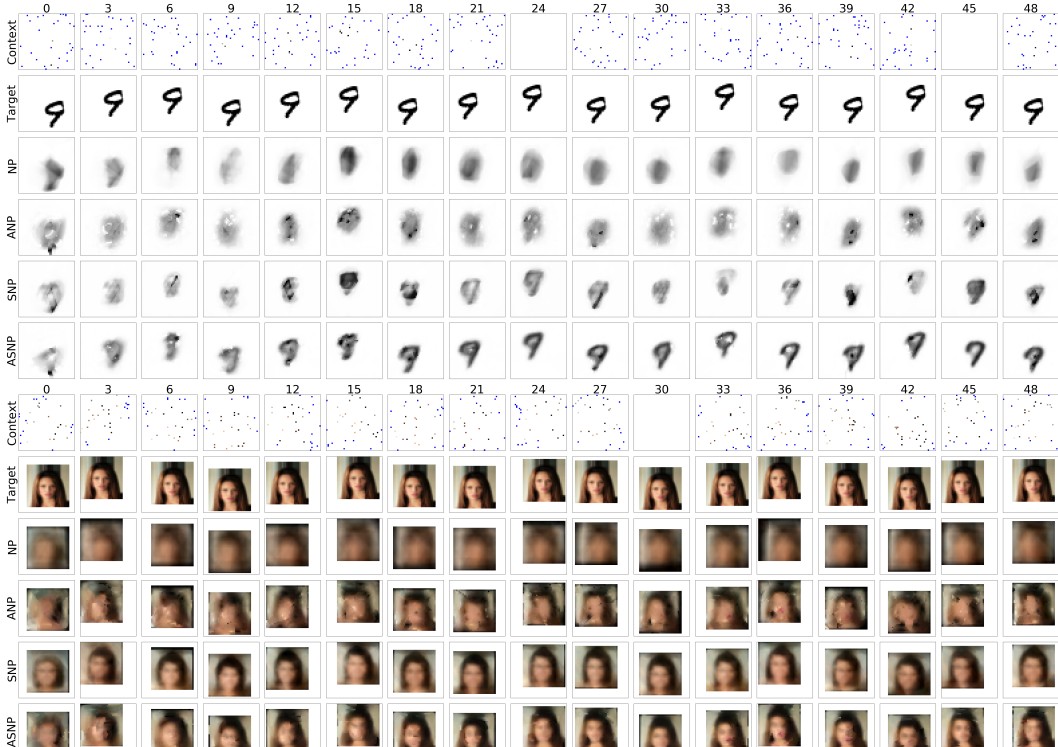

**Figure 5:** 2D moving MNIST (**Top**) and CelebA (**Bottom**) regression samples from scenario (c). In each Context row, white pixels at the background are drawn as blue to visualization.

**2D Regression:** We take a step further to evaluate our model on a more challenging 2D setting where the queries are the pixel locations and the regression is over the pixel values of images. The image is a $42 \times 42$-sized black canvas with a moving MNIST digit (sized $28 \times 28$) or a cropped CelebA face (resized to $32 \times 32$). The digits or the faces start moving from a random location heading towards a direction randomly selected at $t = 0$ with a speed of 3 pixels per time-step and bouncing perfectly upon encountering a wall. Like in the regression task, a small Gaussian noise is added to the location obtained after each transition. Same as the regression task, 3 context regimes are designed. In regime (a) and (b), $n$ is either 0 or $n \in [5, 500]$ while $m$ is drawn from $[1, 501 - n]$. In (c), $n$ is either 0 or 30 and $m$ is drawn from $[1, 51 - n]$. The target set subsumes the context.

We present the quantitative results for 2D regression tasks in Fig. 4. From these, we infer the following points. We observe that SNP performs better than ANP except in the early time-steps of scenario (a) when a large number of contexts points are shown without interruptions. We reason that in the presence of such ample context points, NP and SNP still suffer from under-fitting while ANP can make an effective use of those points without having to model the dynamics. But when the task setting demands that the model capture the dynamics, as in task (c), both NP and ANP perform poorer than SNP and ASNP. This is more pronounced in the 2D setting than in the 1D because tracking dynamics of images without explicit temporal modeling is harder. We further note that MNIST digits are simpler than the CelebA faces. Therefore in the latter case, SNP struggles with underfitting and ANP struggles with modeling the dynamics, causing them both to furnish similar performances in task (b). ASNP does not suffer from either of these problems and it outperforms all baselines in all the tasks. In scenario (a), we also note a sudden performance drop in NP, ANP and ASNP. While unsurprising in NP and ANP, it also occurs in ASNP due to sudden switch-over to a purely imaginary context. This drop does not appear in the 1D tasks since the number of context points provided are fewer and therefore ASNP is less impacted by their absence.

We also evaluate the effect of varying the imaginary context sizes on the moving MNIST data set for the scenarios (a) and (b). We observe that a larger $k$ yields better performance. This gain, however, saturates quickly as the improvement between $k = 9$ and $k = 100$ is not large. This shows that the imaginary context can represent the stochastic process with only a few points.

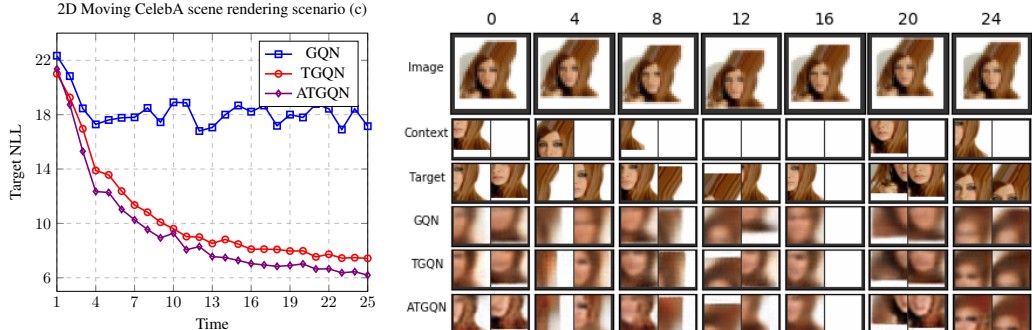

**Figure 6: Left:** Target NLL on moving CelebA 2D scene rendering task for scenario (c) and **Right:** examples.

In Fig. 5, we present the qualitative results on the two data sets for scenario (c). Here, we note that NP and ANP can produce the digit location but not the appearance. SNP captures the appearance of the digit but not clearly. Although ASNP also produces fuzzy appearance in the early time-steps, the digit becomes identifiable by $t = 10$ and the details keep getting clearer over $t = 20$. We also see a similar trend for the moving CelebA task. More illustrations of scenarios (a), (b) and (c) are included in Appendix E.2.

**2D Scene Rendering.** We also introduce the attention paradigm of ASNP into Temporal GQN (Singh et al., 2019) and test it on the moving CelebA rendering task. We call it Attentive TGQN (ATGQN). To design this, we use the Temporal-ConvDRAW as in TGQN (Singh et al., 2019) for encoding a global latent. For computing a query-dependent representation, we apply the ASNP's attention encoder with imagination to get a scene-wise matrix. We use the same decoder as in TGQN. While it is a limited solution working on the task where $y_t$ can be divided as points with queries, it resolves the underfitting for overlapping pixels between the context and the target. Unlike NPs, partial overlap exists, which is difficult to predict by scene-wise attention. More details about the model architecture is described in Appendix B.2.

The canvas size of the data set is $80 \times 80$ and cropped face image size is resized in $64 \times 64$. The direction of motion and initial position are randomly selected at $t = 0$ with 13 pixels per time-step speed. The size of output $y$ is $32 \times 32$. The sequence length is 25. One context view is provided to the model at each of the randomly chosen 20 time-steps out of 25 time-steps. and an empty set is given on the remaining 5 time-steps. The target size $m$ is drawn in the range $[1, 11 - n]$.

In Fig. 6, we show quantitative and qualitative results for scenario (c). We can see that ATGQN outperforms TGQN because for the overlapping areas, ATGQN resolves the underfitting. The performance gap between TGQN and ATGQN, however, is smaller compared to the previously seen 1D and 2D regression tasks. A similar trend is observed in scenario (a) whose results are in Appendix D.4. The reason for this is low uncertainty in the presence of the context that provides a wide view of the canvas. This and the small number of context causes lesser underfitting than what would have arisen had the context observations been highly partial as in the regression tasks. More examples are included in Appendix E.3.

## 6 CONCLUSION

In this paper, we addressed the problem of under-fitting that affects Neural Processes and Sequential Neural Processes. Although this is resolved in the former by ANP, it is not possible to resolve it sufficiently in SNP with direct application of attention on context history. We introduced Attentive Sequential Neural Processes which comprises a novel memory mechanism of imaginary context to resolve the under-fitting. It not only compresses the past knowledge but does it in a way that is geared towards making good predictions with low uncertainty. In the experiments, the proposed model shows superior performance on various tasks for a number of sequential scenarios. Since ASNP is a model based on attention, its limitations include those of any attention-based model. One such case is the scene-wise attention. Scene-wise attention cannot fit properly for partial overlapped region between context and target. Although we partially solve it with pixel-wise attention, it would still be interesting to encode query dependent representation for scene observation.

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

## APPENDIX A    ELBO DERIVATIONS

$\log P(Y|X, C)$

$= \log \mathbb{E}_{Q(Z,C^i|C^r,D)} \left[ \dfrac{P(Y, Z, C^i|X, C^r)}{Q(z, C^i|C^r, D)} \right]$

$= \log \mathbb{E}_{Q(Z,C^i|C^r,D)} \left[ \displaystyle\prod_{t=1}^{T} \dfrac{P(y_t|x_t, z_t, C_t)P(z_t|z_{<t}, C_t)P(U_t^i|X_t^i, C_{<t}, C_t^r)P(X_t^i|C_{<t}, C_t^r)}{Q(z_t|z_{<t}, C_t^i, C^r, D)Q(U_t^i|X_t^i, C_{<t}^i, C^r, D)Q(X_t^i|C_{<t}^i, C^r, D)} \right]$

$\geq \mathbb{E}_{Q(Z,C^i|C^r,D)} \left[ \log \displaystyle\prod_{t=1}^{T} \dfrac{P(y_t|x_t, z_t, C_t)P(z_t|z_{<t}, C_t)P(U_t^i|X_t^i, C_{<t}, C_t^r)P(X_t^i|C_{<t}, C_t^r)}{Q(z_t|z_{<t}, C_t^i, C^r, D)Q(U_t^i|X_t^i, C_{<t}^i, C^r, D)Q(X_t^i|C_{<t}^i, C^r, D)} \right]$

$= \mathbb{E}_{Q(Z,C^i|C^r,D)} \displaystyle\sum_{t=1}^{T} \left[ \log \dfrac{P(y_t|x_t, z_t, C_t)P(z_t|z_{<t}, C_t)P(U_t^i|X_t^i, C_{<t}, C_t^r)P(X_t^i|C_{<t}, C_t^r)}{Q(z_t|z_{<t}, C_t^i, C^r, D)Q(U_t^i|X_t^i, C_{<t}^i, C^r, D)Q(X_t^i|C_{<t}^i, C^r, D)} \right]$

$= \mathbb{E}_{Q(Z,C^i|C^r,D)} \displaystyle\sum_{t=1}^{T} [\log P(y_t|x_t, z_t, C_t) - \log \dfrac{Q(z_t|z_{<t}, C_t^i, C^r, D)}{P(z_t|z_{<t}, C_t)} - \log \dfrac{Q(U_t^i|X_t^i, C_{<t}^i, C^r, D)}{P(U_t^i|X_t^i, C_{<t}, C_t^r)}$

$\qquad - \log \dfrac{Q(X_t^i|C_{<t}^i, C^r, D)}{P(X_t^i|C_{<t}, C_t^r)}]$

$= \displaystyle\sum_{t=1}^{T} \mathbb{E}_{\prod_{t'=1}^{t} Q(z_{t'}|z_{<t'}, C_{t'}^i, C^r, D)Q(C_{t'}^i|C_{<t'}^i, C^r, D)} [\log P(y_t|x_t, z_t, C_t)]$

$\qquad - \mathbb{E}_{\prod_{t'=1}^{t} Q(z_{t'}|z_{<t'}, C_{t'}^i, C^r, D)Q(C_{t'}^i|C_{<t'}^i, C^r, D)} \left[ \log \dfrac{Q(z_t|z_{<t}, C_t^i, C^r, D)}{P(z_t|z_{<t}, C_t)} \right]$

$\qquad - \mathbb{E}_{\prod_{t'=1}^{t} Q(U_{t'}^i|X_{t'}^i, C_{<t'}^i, C^r, D)Q(X_{t'}^i|C_{<t'}^i, C^r, D)} \left[ \log \dfrac{Q(U_t^i|X_t^i, C_{<t}^i, C^r, D)}{P(U_t^i|X_t^i, C_{<t}, C_t^r)} \right]$

$\qquad - \mathbb{E}_{\prod_{t'=1}^{t} Q(X_{t'}^i|C_{<t'}^i, C^r, D)} \left[ \log \dfrac{Q(X_t^i|C_{<t}^i, C^r, D)}{P(X_t^i|C_{<t}, C_t^r)} \right]$

$= \displaystyle\sum_{t=1}^{T} \mathbb{E}_{\prod_{t'=1}^{t} Q(z_{t'}|z_{<t'}, C_{t'}^i, C^r, D)Q(C_{t'}^i|C_{<t'}^i, C^r, D)} [\log P(y_t|x_t, z_t, C_t)]$

$\qquad - \mathbb{E}_{Q(C_t^i|C_{<t}^i, C^r, D)\prod_{t'=1}^{t-1} Q(z_{t'}|z_{<t'}, C_{t'}^i, C^r, D)Q(C_{t'}^i|C_{<t'}^i, C^r, D)} \mathbb{KL}(Q(z_t|z_{<t}, C_t^i, C^r, D) \parallel P(z_t|z_{<t}, C_t))$

$\qquad - \mathbb{E}_{Q(X_t^i|C_{<t}^i, C^r, D)\prod_{t'=1}^{t-1} Q(U_{t'}^i|X_{t'}^i, C_{<t'}^i, C^r, D)Q(X_{t'}^i|C_{<t'}^i, C^r, D)} \mathbb{KL}(Q(U_t^i|X_t^i, C_{<t}^i, C^r, D) \parallel P(U_t^i|X_t^i, C_{<t}, C_t^r))$

$\qquad - \mathbb{E}_{\prod_{t'=1}^{t-1} Q(X_{t'}^i|C_{<t'}^i, C^r, D)} \mathbb{KL}(Q(X_t^i|C_{<t}^i, C^r, D) \parallel P(X_t^i|C_{<t}, C_t^r))$

$= \displaystyle\sum_{t=1}^{T} \mathbb{E}_{\prod_{t'=1}^{t} Q(z_{t'}|z_{<t'}, C_{t'}^i, C^r, D)Q(C_{t'}^i|C_{<t'}^i, C^r, D)} [\log P(y_t|x_t, z_t, C_t)]$

$\qquad - \mathbb{E}_{Q(C_t^i|C_{<t}^i, C^r, D)\prod_{t'=1}^{t-1} Q(z_{t'}|z_{<t'}, C_{t'}^i, C^r, D)Q(C_{t'}^i|C_{<t'}^i, C^r, D)} \mathbb{KL}(Q(z_t|z_{<t}, C_t^i, C^r, D) \parallel P(z_t|z_{<t}, C_t))$

$\qquad - \mathbb{E}_{\prod_{t'=1}^{t-1} Q(C_{t'}^i|C_{<t'}^i, C^r, D)} \mathbb{KL}(Q(C_t^i|C_{<t}^i, C^r, D) \parallel P(C_t^i|C_{<t}^i, C_t^r))$

$= \displaystyle\sum_{t=1}^{T} \mathbb{E}_{\prod_{t'=1}^{t} Q(z_{t'}|z_{<t'}, C_{t'}^i, C^r, D)Q(C_{t'}^i|C_{<t'}^i, C^r, D)} [\log P(y_t|x_t, z_t, C_t)]$

$\qquad - \mathbb{E}_{\prod_{t'=1}^{t-1} Q(z_{t'}, C_{t'}^{i'}|z_{<t'}, C_{<t'}^i, C^r, D)} \mathbb{KL}(Q(z_t, C_t^i|z_{<t}, C_{<t}^i, C^r, D) \parallel P(z_t, C_t^i|z_{<t}, C_{<t}^i, C_t^r))$

$= \displaystyle\sum_{t=1}^{T} \mathbb{E}_{Q_\phi(z_t, C_t^i|C^r, D)} [\log P_\theta(y_t|x_t, z_t, C_t^i, C_t^r)]$

$\qquad - \mathbb{E}_{Q_\phi(z_{<t}, C_{<t}^i)} [\mathbb{KL}(Q_\phi(z_t, C_t^i|C^r, D) \parallel P_\theta(z_t, C_t^i|z_{<t}, C_{<t}^i, C_t^r))]$

where $Q_\phi(z_t, C_t^i | C^r, D) = \prod_{t'=1}^t Q(z_{t'} | z_{<t'}, C_{t'}^i, C^r, D) Q(C_{t'}^i | C_{<t'}^i, C^r, D)$ and $Q_\phi(z_{<t}, C_{<t}^i) = \prod_{t'=1}^{t-1} Q(z_{t'}, C_t^{i\prime} | z_{<t'}, C_{<t'}^i, C^r, D)$ for simplicity.

## APPENDIX B    MODEL DETAILS

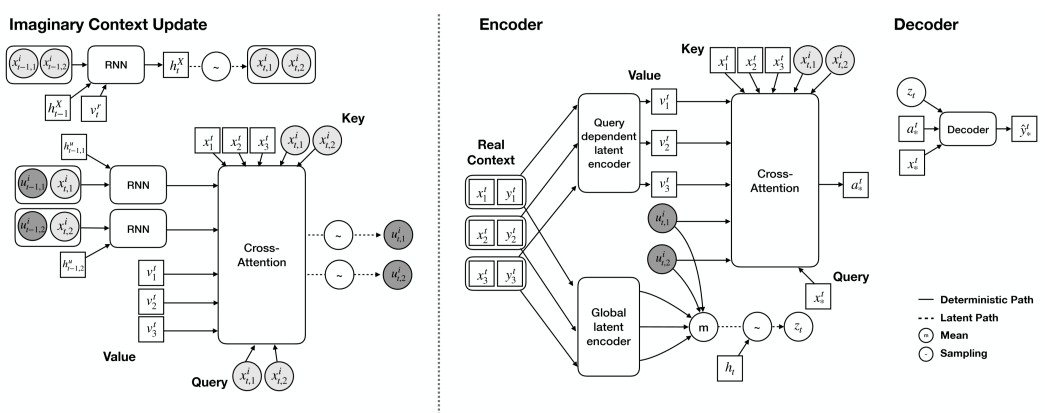

**Figure 7:** Model architecture of Attentive Sequential Neural Processes (ASNP) imaginary context update (**left**) and encoder/decoder (**right**)

### B.1    NPS

Every models share a basic architecture (e.g. context encoder and decoder). They have two encoders. One of them is for a global latent. It consists of 3 layers MLP with ReLU (Nair & Hinton, 2010) is used as encoder and 2 layers MLP is used to encode and sample a latent. Another encoder is consisted of 6 layers MLP with ReLU activation function and mean function for NP and SNP and attention module for ANP and ASNP. Decoder is consisted of 4 layers MLP with ReLU activation function and 1 layer MLP is used to sample $\hat{y}$ with uncertainty.

For non sequential models (NP and ANP), time is encoded as a normalized float scalar, $e_t = 0.25 + 0.5(t/T)$, where $T$ is the length of sequence of data. After encoding, it is appended in query as $x' = (x, e_t)$.

For attention models (ANP and ASNP), Multihead attention (Vaswani et al., 2017) is used because it showed the best performance in a variety of attention methods (Kim et al., 2019). ASNP has two attention modules, which shares parameters.

For sequential models (SNP and ASNP), they uses same temporal architecture for a global latent. LSTM with a default setting of TensorFlow (Abadi et al., 2016) is used and the dimension of hidden unit of LSTM is $d_r$ where $d_r$ is the dimension of representations. For ASNP, two RNN modules for the imaginary queries and representations is LSTM with a default setting of Tensorflow. The dimension of hidden unit of LSTMs is $d_r$. Here, the number of state of LSTM for representations is $k \times$batch size.

The dimension of representation $d_r$ is 128 and the dimension of query $d_q$ is 1 for 1D regression and 2 for 2D regression. Note that for non-sequential model, $d_q$ is 2 and 3 by attaching encoded time $e_t$. The number of imaginary context is 25. The initial imaginary context $C_0^i$ is trainable parameters. Learning rate is 0.0001 and batch size is 16 for 1D regression and 8 for 2D regression. The diagram for ASNP architecture is in Fig. 7.

### B.2    GQNS

For an encoder, tower representation (Eslami et al., 2018) is used. A decoder is similar between models (GQN, TGQN and ATGQN) based on the decoder of CGQN (Kumar et al., 2018) that is a deterministic generative model working on auto-regressive manner. The different is inputs that are described in bellows.

For GQN, time is encoded and appended in query same to NP and ANP. To sample $z_t$, we apply a convolutional DRAW(ConvDRAW) (Gregor et al., 2015; 2016) like GQN (Kumar et al., 2018).

To render $\hat{y}$, the decoder inputs $z_t$, $X_t$ and $v_t^r$ that is an output of the decoder. $v_t^r$ is averaged representation that is an input for every target queries.

For TGQN, Temporal-ConvDRAW (Singh et al., 2019) is applied to sample $z_t$ with ConvLSTM (Xingjian et al., 2015) state, $h_t$. We take RSSM to implement as (Singh et al., 2019). The decoder input is a set of the global latent $z_t$, the target queries $X_t$ and ConvLSTM state $h_t$, in which, $z_t$ and $h_t$ are independent to target queries.

For ATGQN, a global latent is sampled same to TGQN and a query dependent representation of ASNP is used. To make a pixel-wise path, we split scene view as pixel view with queries per each pixel. When many context are given, overlapped between context exists. We leave it to optimize batch operation. We encode the pixel-wise representation with 2 convolutional layers to make same dimension with scene-wise, the dimension of which is [scene-height/4, scene-height/4, 3]. The input of decoder are $z_t$, $X_t$ and $r_t = \{r_i^t\}_{i \in \mathcal{I}(D)}$ that is query dependent representations for each target query $x_i^t$.

The dimension of representation $d_r$ is 128. The number of steps for (Temporal) ConvDRAW to sample $z_t$ is 3 and the dimension of $z$ is 4. Note that $z_t$ is the output of the last roll-out of (Temporal) ConvDRAW. For TGQN and ATGQN, the number of hidden unit for ConvLSTM is 40. The dimension of hidden unit on the decoder is 32 and the number of steps for auto-regressive decoding is 6 and output is cultivated. The query size is 2 (for GQN, it is 3 with $e_t$). The number of pseudo context is 100. Learning rate and batch size is 0.0001 and 4.

## APPENDIX C   GAUSSIAN PROCESS DATA SET

The kernel hyper-parameters, length-scale $l$ and kernel-scale $\sigma$ are selected randomly at $t = 0$ in $[0.7, 1.2]$ and $[1.0, 1.6]$ for scenarios (a) and (b). For scenario (c), $l$ and $\sigma$ are chosen in $[1.2, 1.9]$ and $[1.6, 3.1]$ at $t = 0$. The true underlying dynamics of the kernel hyper-parameters $\Delta l$ and $\Delta \sigma$ are in $[-0.03, 0.03]$ and $[-0.05, 0.05]$ chosen at $t = 0$. Gaussian noise on the dynamics is $\sim \mathcal{N}(0, 0.1)$.

## APPENDIX D   ADDITIONAL EXPERIMENTAL RESULTS

### D.1   LEARNING CURVE ON GP TASK

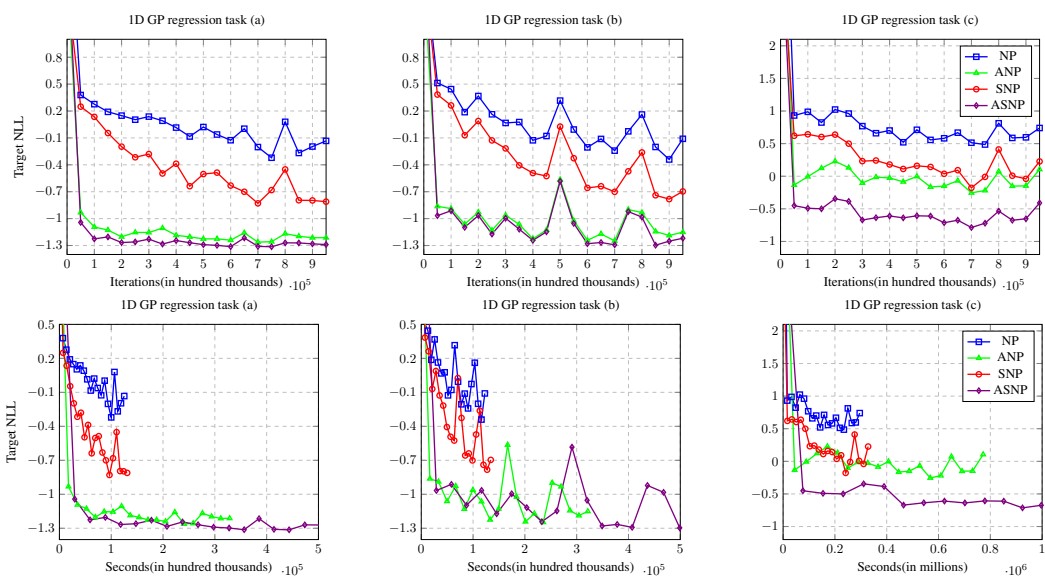

**Figure 8: Top:** Target NLL for iteration. **Bottom:** Target NLL for wall clock time.

In this section, we show target NLL for iterations and wall clock time (Figs. 8) for 1D GP data set. ASNP and ANP computation complexity is bigger than NP $\mathcal{O}(1)$ due to attention. ANP needs $\mathcal{O}(m \sum_t^T n_t)$ where $n_t$ is the context size at time-step $t$. Because it needs to attention to entire context in current. ASNP computation complexity is $\mathcal{O}((n+k)m)$ for generating $r_t$ and $\mathcal{O}((n+k)k)$

for updating the pseudo context. When the sequence of data $T$ is not long, ANP computation complexity is smaller than the complexity of ASNP. However, the learning curve for wall clock time shows ASNP quickly saturates on lower loss in same time.

## D.2 EVALUATION ABOUT BASELINES WITH VARIOUS PARAMETER SIZES FOR GP TASK

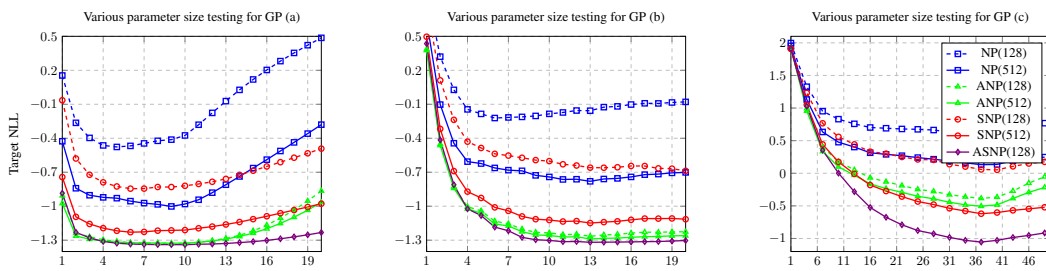

**Figure 9:** Target NLL comparison with baselines having various sizes of $d_r$ (the number in bracket) at each time-step for 1D regression.

In this section, we evaluate our model with baselines implemented with a larger size of neural networks. The representation dimension of baselines (i.e., NP, ANP, and SNP) are selected as $d_r = 512$. This is chosen as the performance improvement saturates beyond this value in (Kim et al., 2019). The remaining settings are the same as described in the previous sections.

Fig. 9 shows that NP and SNP implemented with larger network perform better than their smaller sized counterparts. On the other hand, larger sized ANP shows a similar performance to its smaller counterpart. We thus hypothesize that the smaller model capacity is enough to train ANP on this task. ASNP, although implemented with a smaller network, still clearly outperforms the rest in scenario (c) even against baselines of larger sizes and this gap gets bigger as the time goes on. It shows that the baselines having a bigger network is not sufficient to represent a sequence of stochastic processes and temporal encoding of ASNP and the imaginary context plays a significant role.

## D.3 ASNP WITHOUT IMAGINARY CONTEXT

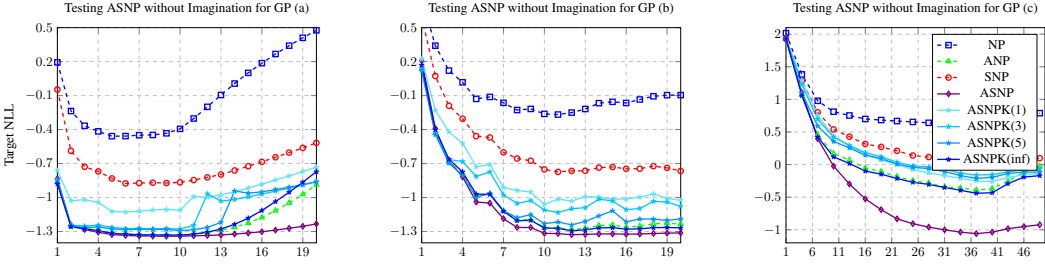

**Figure 10:** Comparison of target NLL of the proposed ASNP versus ASNP without the imaginary context (ASNPK) at each time-step for 1D regression. The number shown in the bracket against ASNPK is $K$, the number of most recent time-steps whose contexts we store for the attention at any given time-step. $K = \inf$ denotes that the entire set of the past contexts are attended on.

In this section, we evaluate the effect of imaginary context on the ASNP performance. To validate it, we design ASNP that does not produce imaginary contexts but simply stores all the real contexts from $K$ most recent time-steps. We call this model as ASNPK in this section. To validate not just the effect of the imaginary context but also the past knowledge on prediction, we test with $K = 1$, 3, 5 and infinity. Here, when $K = $ infinity, ASNPK stores the entire set of the past contexts like ANP and the number of context to attend of ASNPK is usually bigger than the number of imaginary context per each time-step (i.e. if $K = 3$, it could be up to 150 contexts for scenario (a) and (b) while ASNP attends up to 75.). ASNPK has two RNN modules for the global latent and the deterministic variable, but it attends contexts from multiple time-steps. To optimize the attention performance of ASNPK, we also encode the time information into the queries like NP and ANP.

In Fig. 10, quantitative results for the GP tasks are plotted. As $K$ increases from 1 to infinity, AS-NPK shows performance enhancement which quickly saturates. However, it still under-performs

with respect to ANP and ASNP. For ANP, attending all the past knowledge (ANP) makes better performance than what using some past context (ASNPKs except $K$=infinity). ASNPK with $K$ =infinity shows similar performance to ANP because ASNPK(inf) tightly depends on more informative attention than temporal encoding. For ASNP, the simple copy of the past cannot be better than the optimized learned representation of ASNP imaginary contexts. It shows that a sequential memory update mechanism like our brain is more effective than attention on the lossless copies of real past contexts.

### D.4  2D RENDERING TASK FOR SCENARIO (A)

We also test our model on the moving CelebA rendering task for scenario (a). The sequence length $T$ sets as 15 and the context is given to $t = 7$. The context size $n$ is randomly selected in $[1, 10]$ or 0. The target size $m$ is randomly chosen in $[1, 11 - n]$. Other environment setting is same to scenario (c).

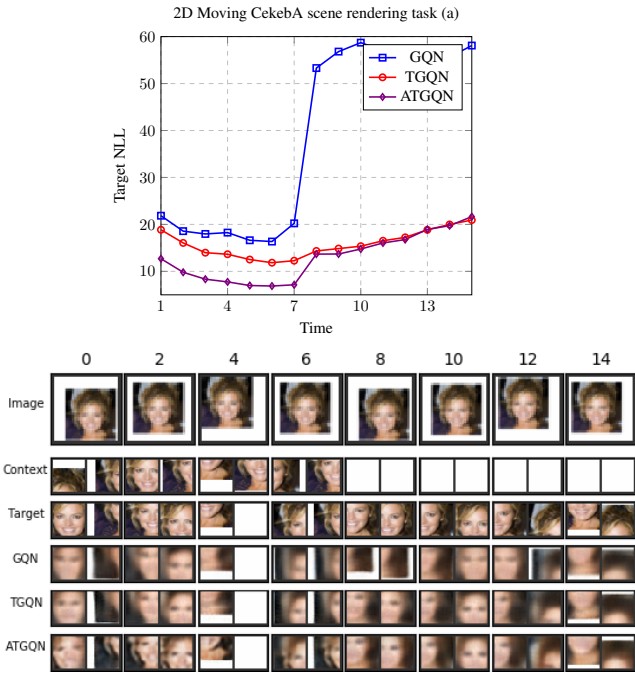

**Figure 11: Top:** Target NLL on moving CelebA 2D scene rendering task for scenario (a) and **Bottom:** examples.

Fig. 11 shows results that are both quantitative and qualitative. Target NLL value of GQN is very high. On the other hand, the generation performance is not the same as the target NLL. The explanation for this is inability to predict the complex face image's location. When context is given, ATGQN outperforms TGQN because ATGQN solves the overlapping region underfitting. ATGQN however shows similar outcomes to TGQN on non-context prediction. The explanation for this is less uncertainty due to a lot of contextual information.

# APPENDIX E    QUALITATIVE RESULTS

## E.1    1D REGRESSION

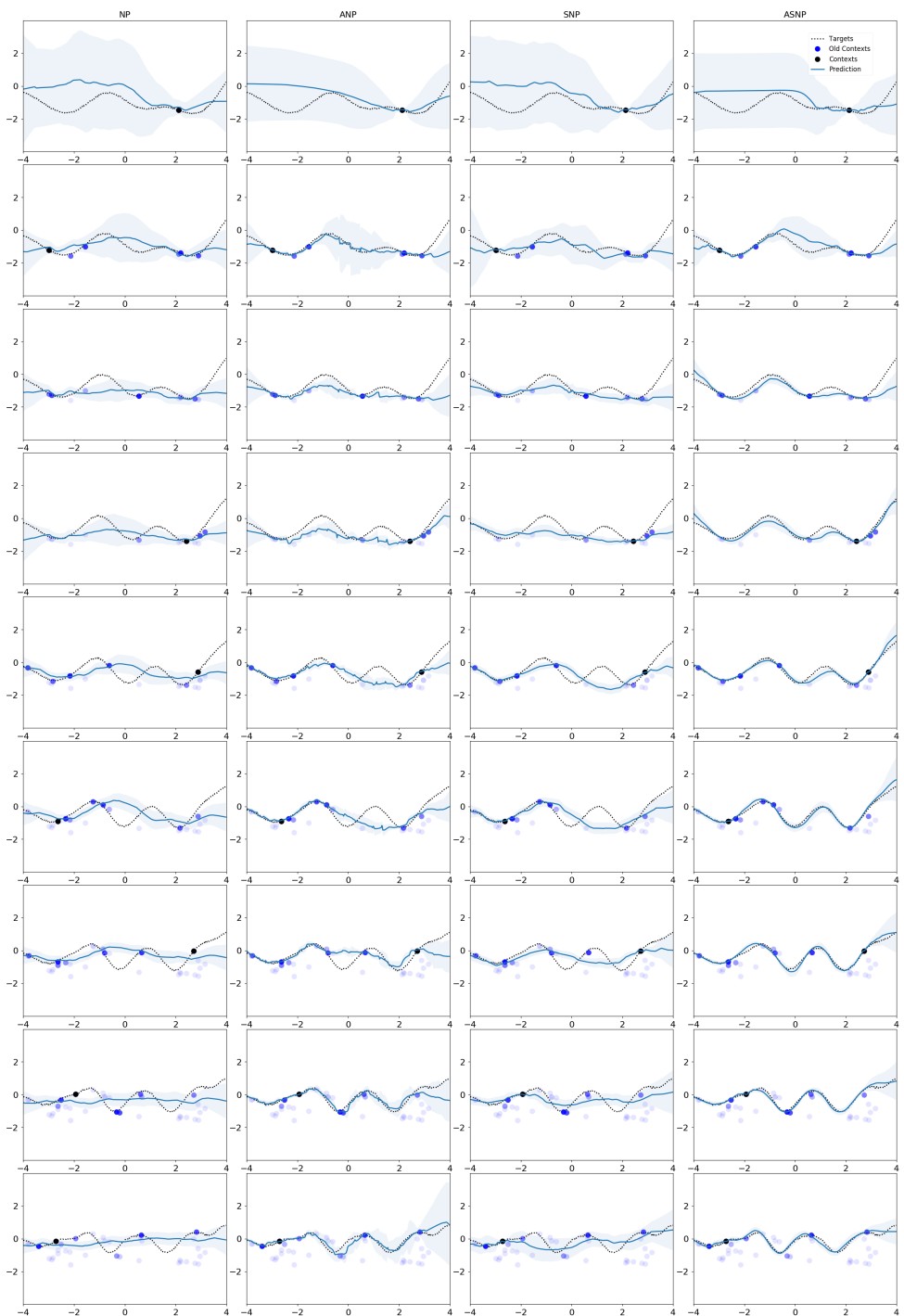

**Figure 12:** 1D GP regression examples for scenario (c). Columns are NP, ANP, SNP and ASNP. Each row is examples at each time-step. Due to space limitations, every $5^{th}$ time-step is shown here instead of every time-step.

## E.2 2D REGRESSION

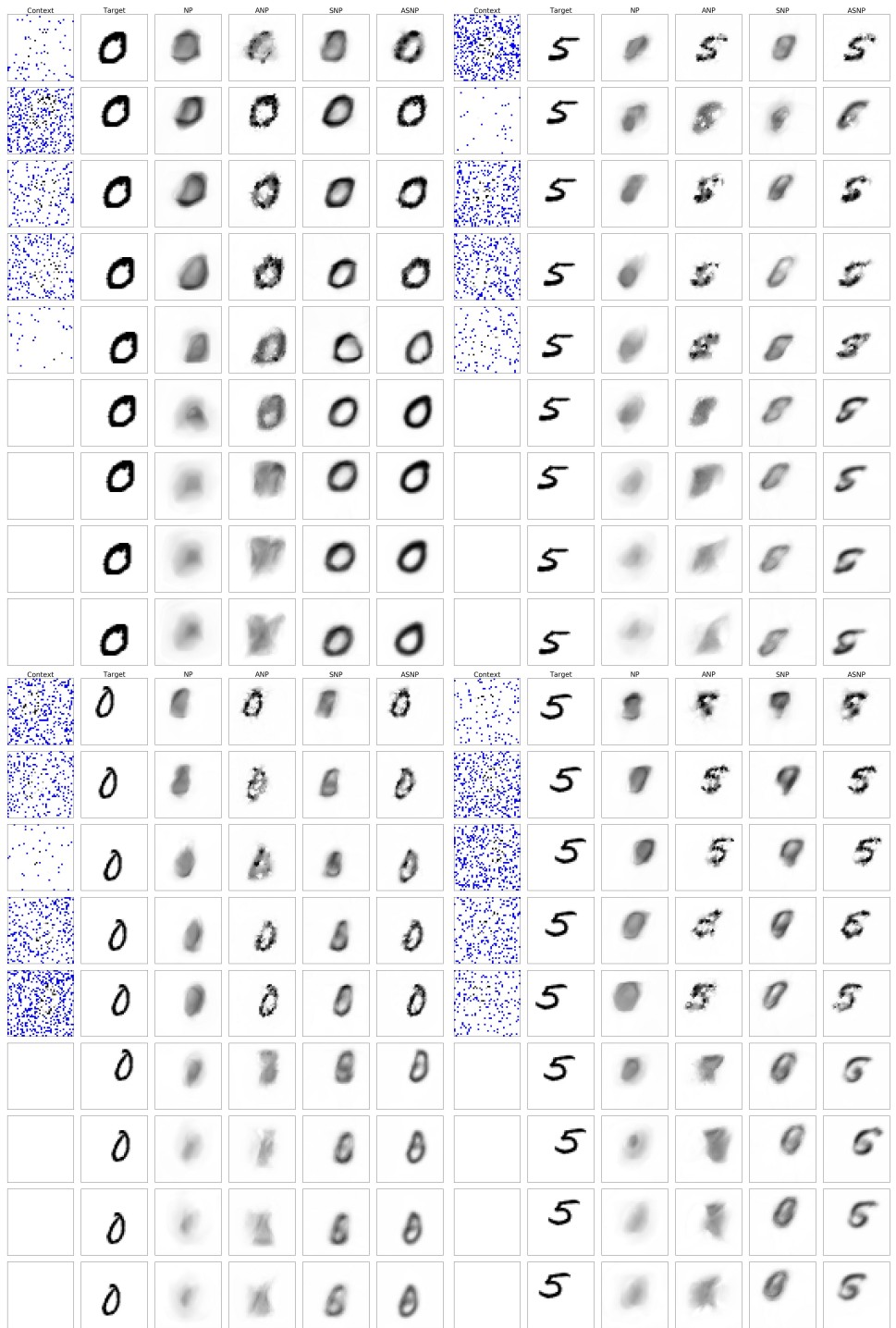

**Figure 13:** 2D moving MNIST regression examples for scenario (a). Columns are Context, Target, NP, ANP, SNP and ASNP. Each row is examples at each time-step. Due to space limitations, every 2[th] time-step is shown here instead of every time-step.

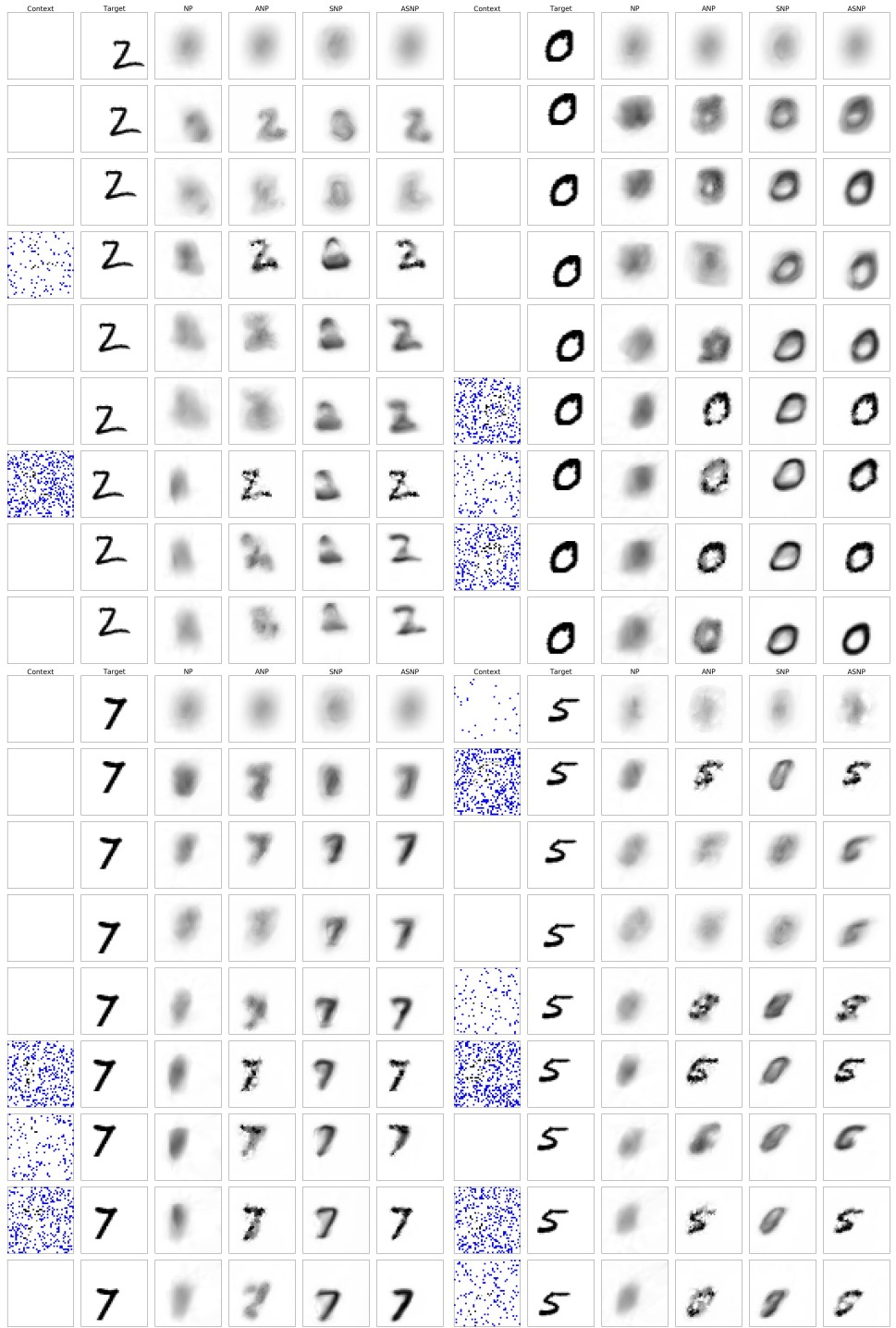

**Figure 14:** 2D moving MNIST regression examples for scenario (b). Columns are Context, Target, NP, ANP, SNP and ASNP. Each row is examples at each time-step. Due to space limitations, every $2^{th}$ time-step is shown here instead of every time-step. It is shown as less than 10 time-steps have context because it doesn't show every time-steps.

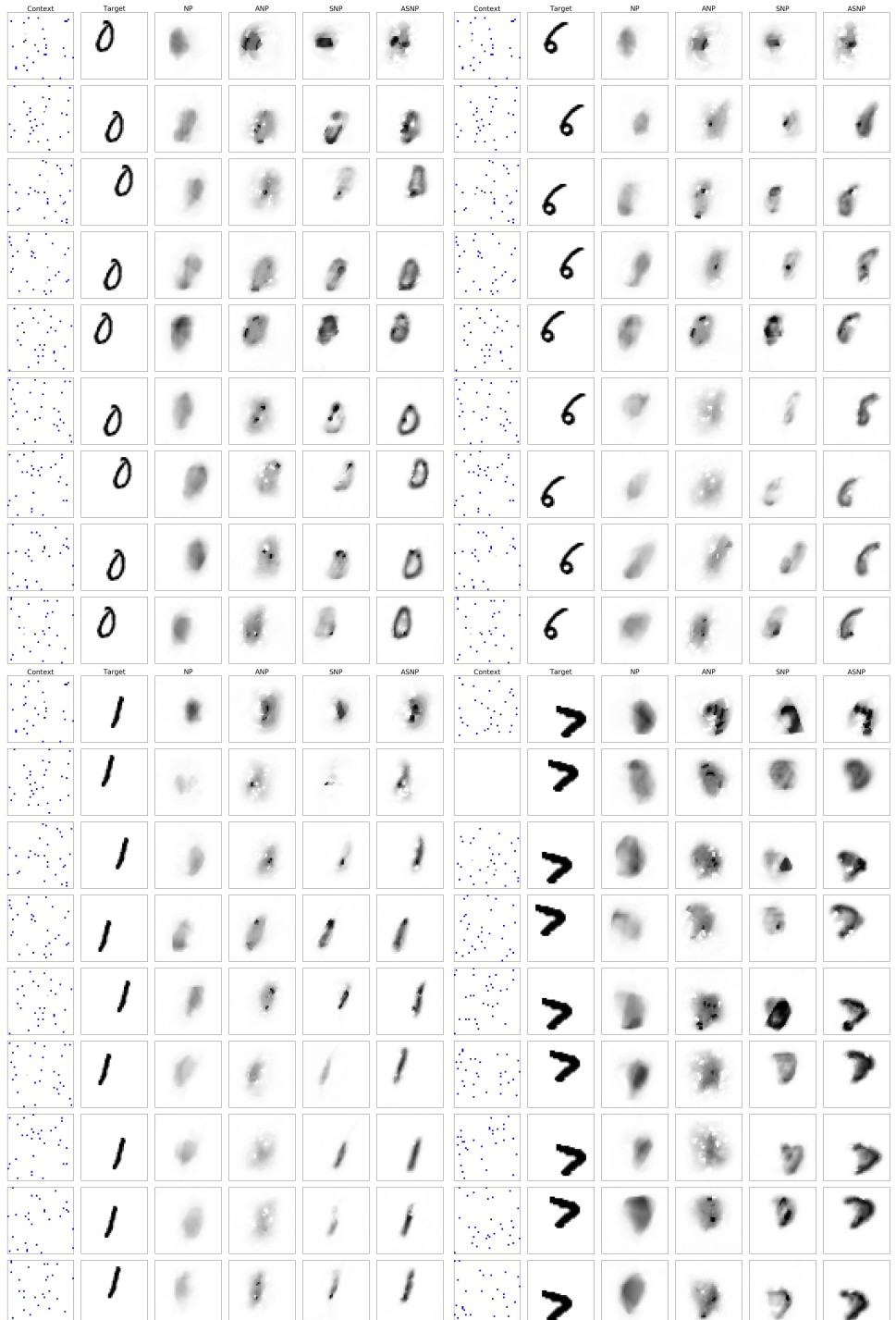

**Figure 15:** 2D moving MNIST regression examples for scenario (c). Columns are Context, Target, NP, ANP, SNP and ASNP. Each row is examples at each time-step. Due to space limitations, every $5^{th}$ time-step is shown here instead of every time-step.

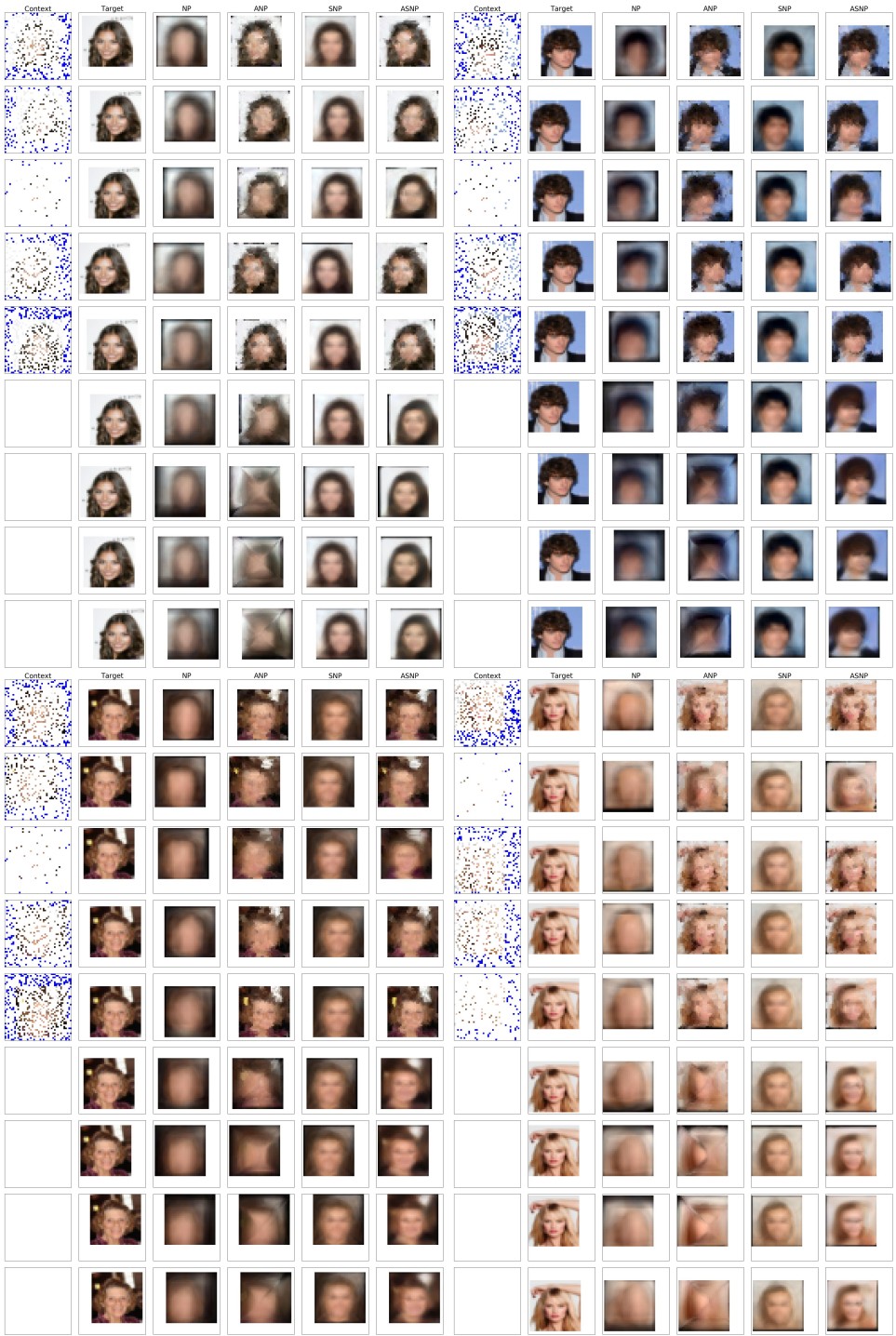

**Figure 16:** 2D moving CelebA regression examples for scenario (a). Columns are Context, Target, NP, ANP, SNP and ASNP. Each row is examples at each time-step. Due to space limitations, every 2$^{\text{th}}$ time-step is shown here instead of every time-step.

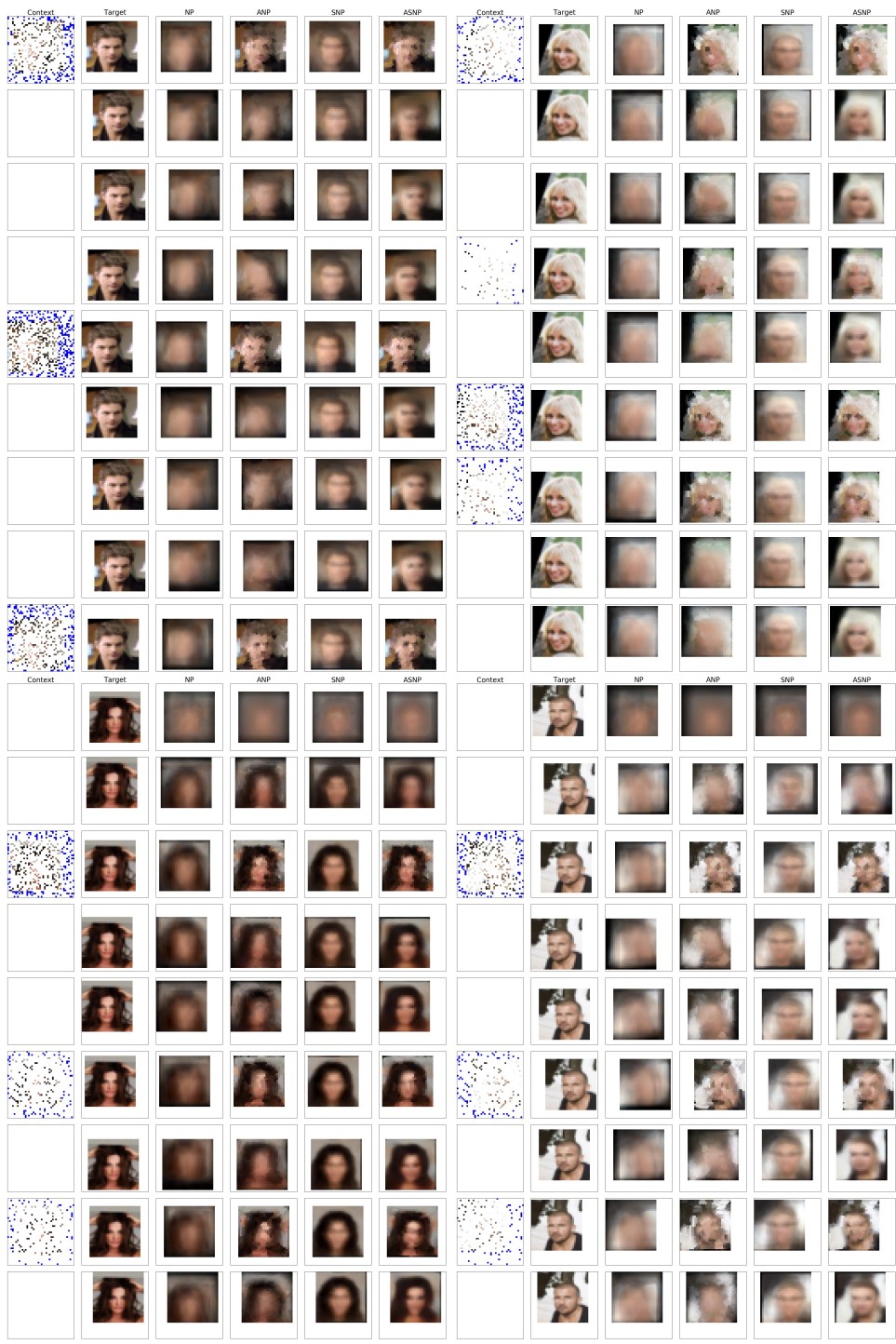

**Figure 17:** 2D moving CelebA regression examples for scenario (b). Columns are Context, Target, NP, ANP, SNP and ASNP. Each row is examples at each time-step. Due to space limitations, every $2^{th}$ time-step is shown here instead of every time-step. It is shown as less than 10 time-steps have context because it doesn't show every time-steps.

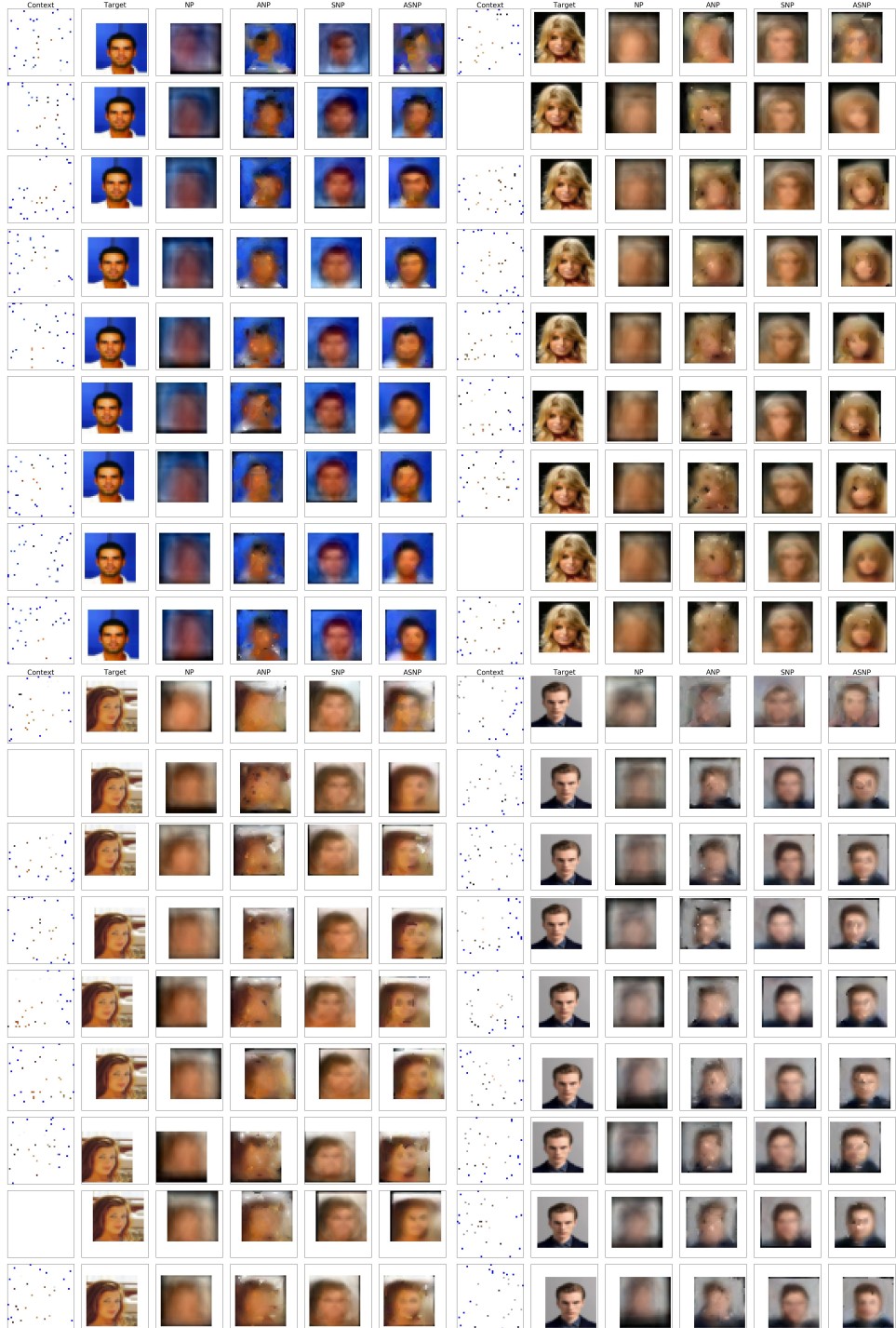

**Figure 18:** 2D moving CelebA regression examples for scenario (c). Columns are Context, Target, NP, ANP, SNP and ASNP. Each row is examples at each time-step. Due to space limitations, every $5^{th}$ time-step is shown here instead of every time-step.

## E.3 2D SCENE RENDERING

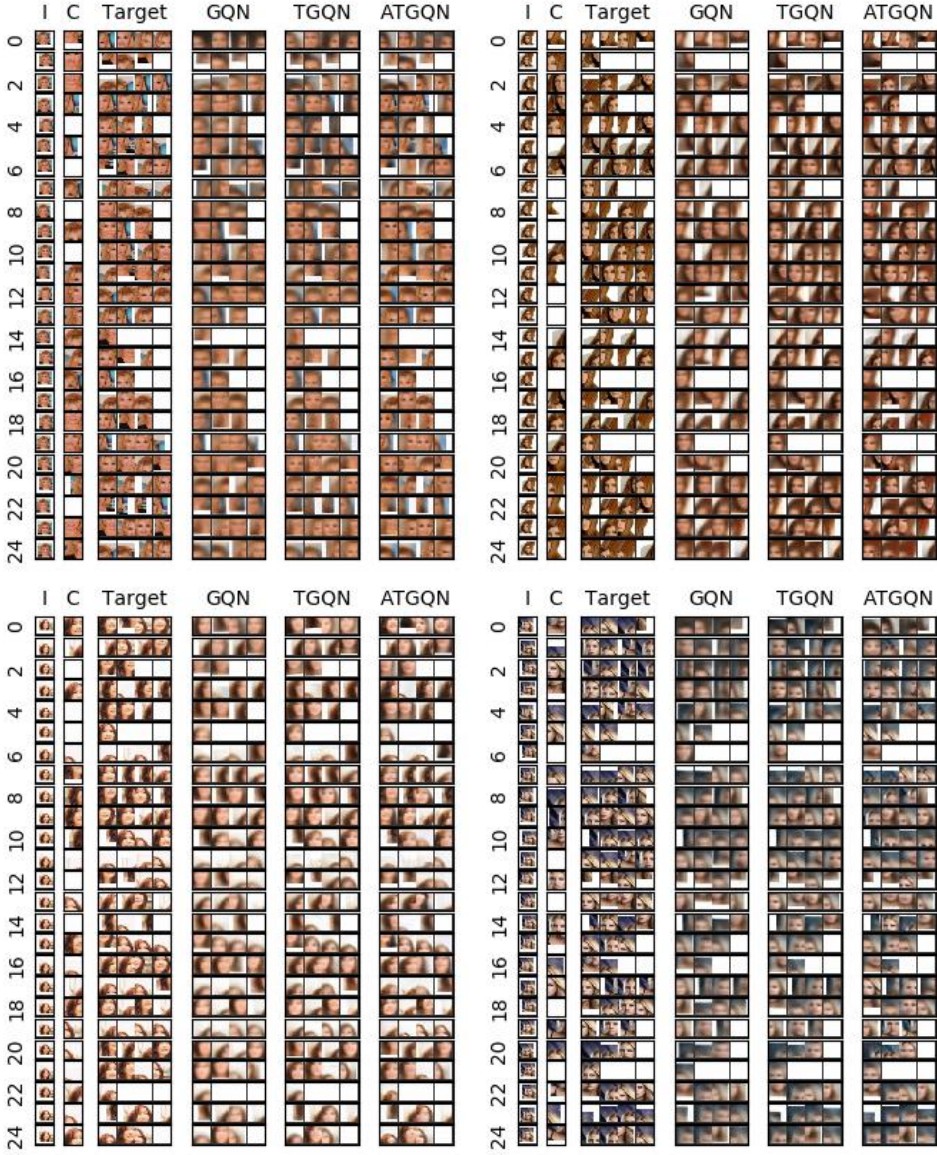

**Figure 19:** moving CelebA 2D scene rendering task examples for scenario (c). Columns are Image (I), Context (C), Target, GQN, TGQN, ATGQN. Each row is examples at each time-step.

