# OpenReview forum: "Attentive Sequential Neural Processes"
_ICLR.cc/2020/Conference — Reject_

### Official Review · AnonReviewer2 · 2019-10-22
**Official Blind Review #2**

**Rating:** 6

**Review:**

Authors present a method to address the problem of underfitting found in sequential neural processes. They cover the literature appropriately in regards to neural processes and developments pertaining to tackling the underfitting problem by applying an attention mechanism. Although, this has successfully been achieved with Neural Processes, the case is different with sequential neural processes, as they cannot store the past context.
Authors addressed this problem by introducing an attention mechanism and model, i.e. Attentive sequential neural processes, which incorporates a memory mechanism of imaginary context. This imaginary context is generated through an RNN network and are treated as latent variables.
The results presented show some promising improvements over other methods used and more results have been included in the appendix. It would be nice to demonstrate the performance in more challenging tasks as well, however the results presented and the new context-imagination introduced are quite promising indeed.
I have read the rebuttal carefully. I appreciate the extra effort put by the authors to address the issues raised from the other reviewers. I think, albeit not ground-breaking research, it could be a good addition to the programme nonetheless.

**Experience Assessment:**

I have read many papers in this area.

**Review Assessment: Checking Correctness Of Derivations And Theory:**

I assessed the sensibility of the derivations and theory.

**Review Assessment: Checking Correctness Of Experiments:**

I assessed the sensibility of the experiments.

**Review Assessment: Thoroughness In Paper Reading:**

I read the paper at least twice and used my best judgement in assessing the paper.

---

> ### Author Response · Authors · 2019-11-15
> **Response to AnonReviewer2**
>
> Thank you for the positive review. Yes, we agree that demonstrating ASNP on more challenging tasks would be a nice and fruitful future work.

---

### Official Review · AnonReviewer1 · 2019-10-23
**Official Blind Review #1**

**Rating:** 1

**Review:**

This paper combines ideas from attentive and sequential neural processes to incorporate an attention mechanism to the existing sequential neural process, which results in an attentive sequential neural processes framework.

While the idea is somewhat interesting, I think this paper is technically vague and not well-motivated, which makes it hard for me to feel convinced that the problem exists and is non-trivial, and that the proposed solution is significant. Let me elaborate on my thoughts below:

First, the authors stated that SNP is subject to the underfitting problem that plagues NP but it is not clear to me why, in the temporal context of SNP, do we need to focus our attention on past contexts, which are no longer relevant. Could the authors please motivate this with a concrete application scenario? Without a concrete scenario, I do not feel very convinced that the problem exists.

Second, the argument that augmenting SNP with an attention mechanism is not trivial is somewhat contrived. In particular, the reason for this non-triviality is that (in the authors' own words) SNP assumes that it cannot store the past context as is -- so what if we simply store the past context & condition the representation on the entire history of past context instead?

Apparently, this can come across trivially by replacing C_t with both C_<t and C_t in Eq. (2). This is in fact very similar to what the authors did in Eq. (4) which summarizes the generative process of ASNP -- the only difference is the generation of imaginary contexts, whose necessity is again questionable, as I elaborate next.

Third, the motivation for imaginary context is pulled from a very distant literature on how a human brain memorizes past experiences in a lossy memory consolidation, which only retains the most important sketches. In the context of ASNP, it is not, however, clear to me why this mechanism is necessary given that entire lossless memory can be stored except that without a lot of contexts, there is not a need for an attention component (as implied in first paragraph of Section 3) which is a contrived motivation.

Fourth, the technical exposition of this paper is too vague. Given that the key contribution here is about an attention component, the background review on ANP is surprisingly informal with no technical detail at all. For the other parts, the technical part is also mostly abstracted away -- what is presented is therefore not that much different from a typical generative model with latent variables, which makes it unclear whether there is a technical challenge here.

In fact, from what I see, going from Eq. (2) to Eq. (4) is not much of a conceptual challenge and the execution of Eq. (4) (particularly the attention component described in Section 3.2) seems like a bunch of arbitrary engineering ideas which were put together to substantiate Eq. (4).

Is there a technical challenge in the entire pipeline that should have been highlighted?

For the experiment, could the author compare the performance between ASNP and ASNP without the imaginery component (but with the attention mechanism)? It would be a good experiment to see if the imaginery component is necessary.

To summarize, I believe the paper in its current state is not well-motivated and appears very incremental given the prior works of SNP and ANP. Even its imaginery component, which is the key contribution here,  is, if I understand Eq. (3) correctly, not much different from context sampling of a NP.


**Experience Assessment:**

I do not know much about this area.

**Review Assessment: Checking Correctness Of Derivations And Theory:**

I assessed the sensibility of the derivations and theory.

**Review Assessment: Checking Correctness Of Experiments:**

I assessed the sensibility of the experiments.

**Review Assessment: Thoroughness In Paper Reading:**

I read the paper at least twice and used my best judgement in assessing the paper.

---

> ### Author Response · Authors · 2019-11-15
> **Response to AnonReviewer1**
>
> Thank you for the detailed review. With these points, we have revised our paper in numerous ways. We address the raised questions in the following points.
>
> Why attend past contexts:   SNP and ASNP are meta-transfer learning frameworks that require fewer observations from the current contexts because they also simultaneously use the information learned in the past. Although the contexts of the past come from a different stochastic process, they are still related to the current stochastic process through the underlying transition dynamics. For instance, consider an agent playing soccer. Its sight is focussed on the ball in the front gathering only a limited observation in the current moment. But using the past knowledge, the player still maintains a dynamic representation of the entire field and especially of the important information (like the locations of the key players) which is useful for making predictions/actions. In the additional experiment (see the response to the next question) where we allow SNP to only attend its own time-step (i.e. K=1) and then increase the K to allow it to attend the past, its performance, not surprisingly, improves with increasing K (although still underperforming against ASNP). So attending to the past contexts is useful.
>
> Why not attend on the entire history of contexts:  We hypothesize and also empirically show that it is a better design choice to have a sequentially updated memory than a simple memory buffer that stores all the observed context points. A sequentially updated memory has the benefit that the model learns to optimize the memory contents for its usefulness in predictions. Another benefit is that it requires fewer storage locations as it does not naively store each and every incoming context point. As mentioned, we compared the proposed ASNP against SNP endowed with attention on lossless memory of all context points gathered in the most recent K time-steps. Although the performance of the latter improves with increasing K = 1 -> 3 -> 5 -> infinity, it quickly saturates at infinity while still under-performing the proposed ASNP clearly highlighting the benefits of the imaginary context. Another interesting point is that when K=infinity, the lossless memory buffer can collect up to 100 or more context points while ASNP outperforms this by attending only on 25 imaginary context points at any given time-step -- clearly highlighting that the imagined context is more size-efficient.
>
> Why analogy to the human brain and need for sequentially-updated memory:   We have reduced the emphasis on the brain analogy in the updated manuscript. Our work is inspired by the under-fitting in SNP that hinders its wider usage. The analogy to the human brain supports our hypothesis that an imagination process for recalling the past is effective and the right way forward to resolve under-fitting.
>
> Technical exposition is too vague:   Thank you for pointing out. We have worked on making the ANP description and the technical exposition of ASNP clearer (see updated manuscript). At the same time, due to space limitations, the finer details have been delegated to the appendices.
>
> The arbitrariness of the design choices: The main idea is in introducing the imaginary context via $P(C_t^i | C_{<t}, C^r_t)$ and our implementation design choices realize that idea -- demonstrated by our better performance on a variety of tasks. The finer design choices are a result of empirical model selection but some broad design choices were hypothesized as follows. A. Imaginary queries should complement the available real context and therefore should depend on them.
> B. Having imagination-tracker RNNs should be beneficial for prediction using the inferred knowledge of the underlying dynamics.
> C. Attention on the tracker RNN hidden states helps capture the pairwise interactions between the context points and also updates the imagined memory with the more correct information from the real contexts.
>
> Key technical challenge:    As responded in a previous question, making use of the past contexts is necessary for the attention to operate on. Realizing that a simple memory buffer of the context history is a sub-optimal design choice, developing the idea and the implementation of the imaginary context was the key technical challenge.
>
> Attention without imaginary context:   As answered in an earlier question, it is possible to truncate the stored contexts to hold the K most recent ones. We have also tested this with K=infinity for the 1D regression tasks as described in the response for all reviewers, we found that our proposed ASNP still outperforms.
>
> NP/ANP not much different from Eq.3:   We politely disagree. Eq.3 depicts how the imaginary queries and values can be propagated from one time-step to the next and it is implemented using attention mechanisms. This is clearly different from NP which does not use attention and also different from ANP because it can neither produce nor propagate imaginary contexts.

---

> > ### Comment · AnonReviewer1 · 2019-11-15
> > **About comparison between ASNP and ASNPK (SNP variant that attends to the entire context history)**
> >
> > Thank you for the extra experiments.
> >
> > Is ASNPK implemented by replacing C_t by C_t and C_<t in the SNP equation -- specifically Eq. (2) in the original version of the manuscript?
> >
> > While the result on the synthetic data looks positive, it would be a lot more convincing if you also show how it would perform in the moving MNIST dataset.
> >
> > I have made a point (see below) earlier on the seemingly trivial technical extension to go from SNP to ASNP and after reading the rebuttal, I am still not very convinced that this extension is non-trivial (since the rebuttal does not elaborate on this point technically) -- could you discuss more on this?
> >
> > "Apparently, this can come across trivially by replacing C_t with both C_<t and C_t in Eq. (2). This is in fact very similar to what the authors did in Eq. (4) which summarizes the generative process of ASNP" -- if you could, please refer to the original version or elaborate if somehow it is no longer relevant.
> >
> > More minor questions:
> >
> > In Figure 10, how many context points are being used per time step for ASNP?
> >
> > In the rebuttal you mentioned that ASNP only generated 25 imaginery context points but in the latest manuscript, you mentioned that it generated 75 context points.
> >
> > Also for ASNPK, is the total number of context points 100 or 150? -- the latest manuscript said 150.

---

> > > ### Author Response · Authors · 2019-11-15
> > > **Response A1.1**
> > >
> > > * “Is ASNPK implemented by replacing C_t by C_t and C_<t in the SNP equation -- specifically Eq. (2) in the original version of the manuscript?”
> > >
> > > Yes, ASNPK is an extended SNP that is also allowed to perform attention on a memory buffer that stores all the context history $C_t$ and $C_{<t}$ as key-value set. We would like to emphasize that vanilla SNP does not have attention mechanism and what we implement as ASNPK uses attention on the context history.
> > >
> > >
> > > * “While the result on the synthetic data looks positive, it would be a lot more convincing if you also show how it would perform in the moving MNIST dataset.”
> > >
> > > With more time until camera ready, we will compute these results. 1D regression is a good representative of the problem setting and we expect the result trend to be similar in 2D settings also.
> > >
> > >
> > > * “ ‘Apparently, this can come across trivially by replacing C_t with both C_<t and C_t in Eq. (2). This is in fact very similar to what the authors did in Eq. (4) which summarizes the generative process of ASNP’ -- if you could, please refer to the original version or elaborate if somehow it is no longer relevant.”
> > >
> > > ASNPK extends SNP in the following way i.e. attending on $C_{\le t}$ rather than just $C_t$. But vanilla SNP does not have attention mechanism so the transformation from SNP to ASNPK is not trivial.
> > >
> > > Furthermore, between ASNPK (which the reviewer seems to refer to by saying Eq.2) and ASNP (which reviewer is referring by saying Eq. 4), we contend against saying that the two are similar. In general, we would like to refrain from reductively bringing the two generative model equations into the same form. In ASNPK, $C_{<t}$ is a static, stale and unoptimized representation of the past stored in a memory. On the other hand, ASNP has $C_t = C_t^i \cup C_t^r$ and the imaginary context is sequentially updated and optimized representation whose transition model is implemented by a different attentive mechanism.
> > >
> > > * “In Figure 10, how many context points are being used per time step for ASNP? In the rebuttal you mentioned that ASNP only generated 25 imaginery context points but in the latest manuscript, you mentioned that it generated 75 context points. Also for ASNPK, is the total number of context points 100 or 150? -- the latest manuscript said 150.”
> > >
> > > The values in the manuscript are more accurate and they override the ones mentioned the previous response. But the key point is that the imaginary context is a more efficient storage in terms of size while performing better than the memory buffer.

---

### Official Review · AnonReviewer3 · 2019-10-25
**Official Blind Review #3**

**Rating:** 3

**Review:**

This paper deals with the underfitting problem happening in neural process and sequential neural process (SNP). The idea is to incorporate the attention scheme in SNP and carry out the so-called attentive sequential neural process (ASNP) for sequence learning.

Strength:
1. A combination of attention into SNP.
2. Some formulations were provided.
3. Different tasks were evaluated to investigate the merit of this method.

Weakness:
1. The comparison for time complexity and parameter size was missing.
2. The labels in figures were inconsistent.
3. An incremental research.

**Experience Assessment:**

I have published one or two papers in this area.

**Review Assessment: Checking Correctness Of Derivations And Theory:**

I did not assess the derivations or theory.

**Review Assessment: Checking Correctness Of Experiments:**

I assessed the sensibility of the experiments.

**Review Assessment: Thoroughness In Paper Reading:**

I made a quick assessment of this paper.

---

> ### Author Response · Authors · 2019-11-07
> **Response to AnonReviewer3**
>
> We are grateful for the comments. We have modified the paper and addressed them as follows.
>
> Time-complexity comparison:   In the appendix, we show the training curves against the wall clock time. It shows that the proposed ASNP converges the fastest among the baselines.
>
> Parameter-size comparison: We tested the proposed ASNP with representation size n=128 against the baselines NP, ANP, and SNP with n=128 and n=512. While the baselines improve going from n=128 to n=512, they still underperform ASNP. In scenario c) which tests how well a model accumulates contexts over time, this gap is clear. The reason we choose size=512 in the baselines is that it shows a similar performance when size=1024 in ANP (Kim et al.) and also needs a shorter training time that allowed us to produce these new findings within the rebuttal period. From these findings, we can say that having a bigger latent size or equivalently more parameters in the baselines is not sufficient and imaginary context itself plays a useful role.
>
> Figure label inconsistency:   We have fixed the inconsistent labels of the figures.
>
> Incremental Research:   The problem we address (as also described in the response for all reviewers) is real and an important one. From this perspective, we believe our performance gains are a significant step forward. NP and SNP are crucial meta-learning frameworks with nice properties which were originally demonstrated on relatively simpler tasks. Addressing under-fitting is the key to making them usable in realistic settings. Our imaginary context shows that attention on a sequentially-updated memory outperforms using a lossless copy of the past while also being more size-efficient.

---

### Author Response · Authors · 2019-11-15
**Global Response**

We thank all the reviewers for their insightful comments. While we also address each reviewer individually, we feel that re-emphasizing our motivation and contributions would orient the reviewers to better assess our work and our responses and to possibly adjust our scores.

* Motivation
SNP is a new class of models that can deal with a broad range of new modeling problems i.e., sequential meta-transfer learning. Given that this model has many useful potential applications, it is an important problem to study the challenges that arise in scaling it to more real and complex settings. Considering that its non-temporal version, NP, suffers from significant under-fitting, an important unanswered question arises with regard to SNP is whether SNP suffers from under-fitting or not (it does) leading to other follow-up questions: if so, how severely? (Significantly). To resolve this, can the existing solutions (like ANP) be directly used? (No). If not, could the naive extension of the existing solution work? (Sub-optimally). Can we do better? (Yes). Our contribution in this paper is thus not simply to apply attention to SNP but to analyze and study the model to answer all of the above questions and propose a model that claims a better modeling hypothesis than the existing contemporary solutions.

* Our Contributions
We provide empirical evidence that underfitting indeed severely deteriorates SNP by showing that the standard SNP significantly improves with our proposed solution on various tasks. We found that without our attention mechanism, SNP can only provide very suboptimal performance. Our work shows not only that attention can resolve this but that it is also not enough to perform attention on a memory buffer that simply stores all the observed contexts. Instead, attention should be performed on a memory that is also sequentially updated. Consequently, a) this memory would learn to store a temporally optimized representation of the past geared towards better prediction and b) this memory would require fewer storage locations as it does not need to naively store each and every observed context point but provides an optimized set of learned memory buffer. And indeed, we empirically show that SNP extended with naive attention on a memory buffer of all the past contexts under-performs the proposed ASNP even if it has both sequential encoding and memory, and that the use of a sequentially ‘imagined’ memory is a better choice. Our experiments also show that fewer storage locations are needed in the imagined memory while furnishing superior performance as compared to the naive memory buffer. Lastly, in a comprehensive set of experiments on 1D and 2D regression and rendering tasks, we demonstrate ASNP's performance gains over NP, ANP, SNP, GQN, and TGQN in different context regimes.

* Additional Experiments:
We would like to highlight some additional experimental results that were computed based on the reviewers’ suggestions that made our claim stronger. Our existing results showed that ASNP outperforms the non-sequential frameworks i.e., NP and ANP and also the sequential one i.e. SNP. While we maintained that our novel imaginary context led to our improved results, reviewers raised some alternative hypotheses which prompted us to experimentally investigate:
    a) Baselines with a larger number of parameters could achieve similar results. (Anon. Reviewer 3)
    b) SNP with attention to the lossless memory of all contexts could achieve similar results. (Anon. Reviewer 1)
These alternatives are put to rest in our new results showing that ASNP still outperforms them. The plots for the new experiments on the 1D regression tasks have been added to the appendix of the updated manuscript. More details about these experiments are also described in the respective responses to the reviewers.

---

### Decision · Program_Chairs · 2019-12-19

**Decision:**

Reject

**Comment:**

This manuscript outlines a method to improve the described under-fitting issues of sequential neural processes. The primary contribution is an attention mechanism depending on a context generated through an RNN network. Empirical evaluation indicates empirical results on some benchmark tasks.

In reviews and discussion, the reviewers and AC agreed that the results look promising, albeit on somewhat simplified tasks. It was also brought up in reviews and discussions that the technical contributions seem to be incremental. This combined with limited empirical evaluation suggests that this work might be preliminary for conference publication. Overall, the manuscript in its current state is borderline and would be significantly improved wither by additional conceptual contributions, or by a more thorough empirical evaluation.